# Rapid custom prototyping of soft poroelastic biosensor for simultaneous epicardial recording and imaging

Bongjoong Kim[1,9], Arvin H. Soepriatna [2,9], Woohyun Park [1], Haesoo Moon[2], Abigail Cox [3], Jianchao Zhao[4], Nevin S. Gupta[4], Chi Hoon Park[4,5], Kyunghun Kim[2], Yale Jeon[2,6], Hanmin Jang [2,6], Dong Rip Kim [6], Hyowon Lee [2], Kwan-Soo Lee [4✉], Craig J. Goergen [2✉] & Chi Hwan Lee [1,2,7,8✉]

The growing need for the implementation of stretchable biosensors in the body has driven rapid prototyping schemes through the direct ink writing of multidimensional functional architectures. Recent approaches employ biocompatible inks that are dispensable through an automated nozzle injection system. However, their application in medical practices remains challenged in reliable recording due to their viscoelastic nature that yields mechanical and electrical hysteresis under periodic large strains. Herein, we report sponge-like poroelastic silicone composites adaptable for high-precision direct writing of custom-designed stretchable biosensors, which are soft and insensitive to strains. Their unique structural properties yield a robust coupling to living tissues, enabling high-fidelity recording of spatiotemporal electrophysiological activity and real-time ultrasound imaging for visual feedback. In vivo evaluations of custom-fit biosensors in a murine acute myocardial infarction model demonstrate a potential clinical utility in the simultaneous intraoperative recording and imaging on the epicardium, which may guide definitive surgical treatments.

---

[1] School of Mechanical Engineering, Purdue University, West Lafayette, IN, USA. [2] Weldon School of Biomedical Engineering, Purdue University, West Lafayette, IN, USA. [3] Department of Comparative Pathobiology, Purdue College of Veterinary Medicine, West Lafayette, IN, USA. [4] Chemical Diagnostics and Engineering, Los Alamos National Laboratory, Los Alamos, NM, USA. [5] Department of Energy Engineering, Gyeongnam National University of Science and Technology, Jinju-Si, Republic of Korea. [6] School of Mechanical Engineering, Hanyang University, Seoul, Republic of Korea. [7] Department of Materials Engineering, Purdue University, West Lafayette, IN, USA. [8] Birck Nanotechnology Center, Purdue University, West Lafayette, IN, USA. [9] These authors contributed equally: Bongjoong Kim, Arvin H. Soepriatna. ✉email: kslee@lanl.gov; cgoergen@purdue.edu; lee2270@purdue.edu

Printed biosensors are integral to the development of various medical research platforms and their broad applications at all scales from cellular to organ level.[1–5] Direct ink writing (DIW) of multidimensional functional architectures on various biological substrates enables rapid prototyping and customization of a range of biosensors with geometrical complexities.[6–8] This approach eliminates the need for multiple steps, masks, and dedicated tools that are typically required in conventional lithography-based techniques.[9–13] Advanced strategies involve the use of conducting polymer inks or silicone composite inks containing conductive nanofillers to serve as dispensable inks for a nozzle injection system that allows for automated rapid prototyping.[14–17] To this end, precise control of the rheological properties of the inks is required in order to (1) ensure high-precision printability for sophisticated rendering at the microscale and (2) prevent the hindering of the densely dispersed nanofillers from the polymerization of the inks.[18] Despite great successes, these inks exhibit both mechanical and electrical hysteresis under periodic large strains due to their viscoelastic nature and/or result in irreversible degradation in conductivity due to the difficulty of maintaining the percolation network of the conductive nanofillers.[19–22] In addition, the viscoelastic inks may produce a risk for delamination from the biocompatible substrates under a large deformation due to the low interaction energy at the interface and the discrepancy in their intrinsic elasticity.[23] The fundamental limitations of these inks impede their implementation in medicine, particularly under conditions that demand reliable recording against repetitive loading such as the cardiac cycle.

Herein, we introduce a sponge-like form of poroelastic silicone composites with optimal rheological properties that allow it to be printed in a nozzle injection system at the microscale. These poroelastic silicone composites provide the following unique features: (1) poroelastic behavior (rather than viscoelastic behavior) with reversible compressibility that can effectively suppress both mechanical and electrical hysteresis against repetitive loading cycles; (2) exceptional softness due to the ultralow mechanical modulus ($E < 30$ kPa) of the sponge-like foam, which is lower than that of commercial dispensable inks ($E > 1.11$ MPa; SE 1700, Dow Corning) by more than 10-fold and comparable to that of human cardiac tissues (29–41 kPa); and (3) reliable structural integrity in which conductive nanofillers are integrated through the internal pores of the sponge-like foam to minimize risk of delamination or separation against cyclic deformations. The comparison of these poroelastic silicone composites with other existing materials in terms of the mechanical and electrical properties is shown in Supplementary Table 1. In this report, we elucidated the structure-property relationships of these poroelastic silicone composites at the molecular and microsystemic levels and then evaluated their applicability in rapid custom prototyping of stretchable biosensors. To demonstrate the utility of this concept in medical practice, we produced a range of custom-fit biosensors tailored for simultaneous recording and imaging of hearts with acute myocardial infarction in vivo.

## Results

**Custom design and prototyping of poroelastic biosensors.** Figure 1 displays schematic illustrations and the corresponding optical images for the fabrication of a poroelastic biosensor array custom fitted to the infarcted area of the heart. The initial design process began by capturing the overall size, geometry, and structure of the infarcted region of the heart through the four-dimensional (4D) segmentation (i.e., 3D geometric volume over a cardiac cycle) of the myocardium via non-invasive ultrasound imaging (Fig. 1a).[24,25] Details of the 4D heart segmentation are included in the Methods section. This 3D geometry was taken into consideration to precisely scale, adjust, and tailor the overall layout of the device to meet the requirement of a specific geometric accuracy. This custom design allowed the recording electrodes of the device to be precisely aligned to the position and orientation of the infarcted area of the heart. Subsequently, the printing process began by directly writing a formulated ink on a Si wafer through the use of an automated nozzle injection system equipped on a three-axis computer-controlled translation stage (Nordson EFD; minimum inner diameter of nozzles = 100 μm; repeatability = ±3 μm; dispensing rate = 20 mm min⁻¹). This printing setting offers the capability to uniformly render micro-scale motifs (thickness ≥ 50 μm-thick; width ≥ 100 μm-wide) that fit into the pre-designed layout of the device.

Herein, the formulated ink, as described in more detail later, contains an optimal mixture of silicone resins and silica particles to provide both high-precision printability and the capability of turning the ink into a sponge-like foam. As presented in Fig. 1b, the selectively patterned traces of the as-printed (i.e., prepolymer) ink were polymerized into an amorphous sponge-like morphology with a pore diameter ranging from 5 to 50 μm under a pressurized hot steam condition of 120 °C and 15 psi using a pressure rice cooker (Max, Instant Pot, Inc). Supplementary Fig. 1a shows representative results of the differential scanning calorimetry (DSC) of the ink to confirm its amorphous character. The formation of the micropores under this condition is likely attributed to the massive penetration and evaporation of the pressurized water molecules to/from the prepolymer.[26] The resulting sponge-like foam was then immersed in a mixture solution of hexane (200 ml) and Ag flakes (0.5 mg; 200 nm–5 μm in diameter; Inframat Advanced Materials, LLC), allowing it to absorb the hexane quickly by capillary action in a manner that traps the Ag flakes into the internal pores (Fig. 1c). The Ag flakes were then plated with Cu for 30 minutes to secure their mechanical and electrical interconnections. The resulting sheet resistance and stretchability were lower than $7.72 \pm 1.52 \, \Omega \, sq^{-1}$ and greater than 100%, respectively (Supplementary Fig. 1b). Here, the Cu-plated Ag flakes were alternatively used as substitutes to expensive Au products, while their entire surface was plated with Au for 2 min to promote biocompatibility.[27] Next, another direct writing of the formulated ink was followed to define the remaining traces of the device in an open mesh layout to ensure breathability and stretchability (Fig. 1d).[28] Finally, a water-soluble film made of polyvinyl alcohol (PVA, Sigma-Aldrich; 50 μm-thick) was applied to gently peel the device from the Si wafer and then trimmed to remove excess areas using scissors (Fig. 1e). The water-soluble film provides excellent biocompatibility and has been used for many implantable medical devices without inflammatory responses.[29,30] Here, the water-soluble film was used as a temporary supporting layer to facilitate easy integration across the surface of the heart under median sternotomy, which thereafter was dissolved within no more than 30 s by applying a warm saline solution. Supplementary Fig. 2a presents a series of images for the complete dissolution of the water-soluble film in 10 s when inserted into a warm water bath (35 °C). Supplementary Fig. 2b provides the corresponding results of computational analysis, revealing that the total bending stiffness (or flexural rigidity) of the device decreased exponentially as the supporting layer (i.e., water-soluble film) was dissolved over time due to its cubic dependence on thickness.[28] This low bending stiffness ($< 8.0 \times 10^7$ GPa μm⁴) could substantially reduce the required minimum adhesion energy per unit area, thereby providing a strong capillary adhesion to the epicardium.[28]

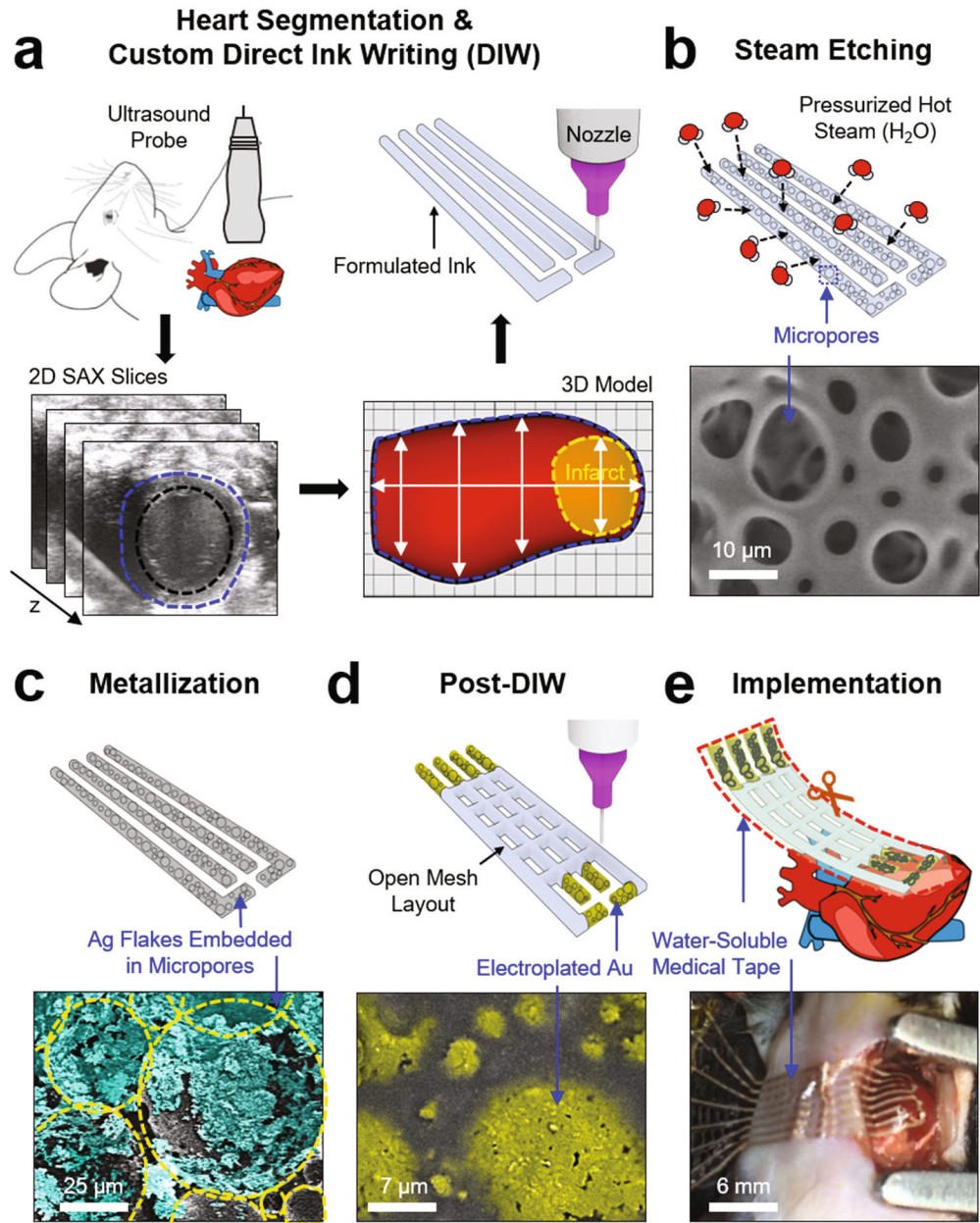

**Fig. 1 Custom design and prototyping of poroelastic biosensor array.** Schematic illustrations and optical images for the key process steps: (**a**) 3D imaging and custom direct ink writing (DIW), (**b**) steam etching, (**c**) metallization, (**d**) post-DIW, and (**e**) implementation on the epicardial surface of a murine heart.

**Material structure-property relationships.** The rheological properties of the formulated ink are key parameters governing the printability and structural integrity of the printed device.[18] To achieve the optimal rheological properties, we formulated the ink by blending (1) a base resin comprised of vinyl terminated diphenylsiloxane-dimethyl silicone copolymer, methylhydrosiloxane copolymer, and siloxane monomer with the weight ratio of 6.5:3.3:0.2, (2) a dilute resin (Sylgard 184, Dow Corning; 10:1 weight ratio of base and curing agents), and (3) a physical cross-linker of polysiloxane-treated hydrophobic silica ($SiO_2$–PS) particles (CAB-O-SIL® fumed silica TS-720, CABOT Corp). To predict the miscibility of the blended inks, we first conducted molecular dynamics (MD) simulations and quantitatively evaluated the interfacial interaction energy between the physical cross-linker (i.e., $SiO_2$–PS particles) and the surrounding resins.

Figure 2a shows a representative snapshot image of the MD simulation for a $SiO_2$-PS particle, compared to a non-treated (hydroxyl-terminated) silica ($SiO_2$–OH) particle as a control. Details of the MD simulation results are summarized in Supplementary Table 2. The results show that the interaction energy of the $SiO_2$-PS particle ($-1.852 \times 10^{21}$ kcal mol$^{-1}$ g$^{-1}$) remained substantially lower than the control $SiO_2$–OH particle ($-6.969 \times 10^{21}$ kcal mol$^{-1}$ g$^{-1}$), implying that the $SiO_2$–PS particle provides enhanced miscibility with the surrounding resins to better serve as a tractable modifier for the rheological properties of the ink.[32]

To characterize rheological properties, we prepared variously blended inks by varying the weight ratio of the base resin, dilute resin, and $SiO_2$-PS particles from 5.7:3.3:1.0 to 6.0:3.3:0.7, 4.2:5.0:0.8, and 4.5:5.0:0.5, compared to commercial control groups of a dispensable silicone ink (SE 1700, Dow Corning) and

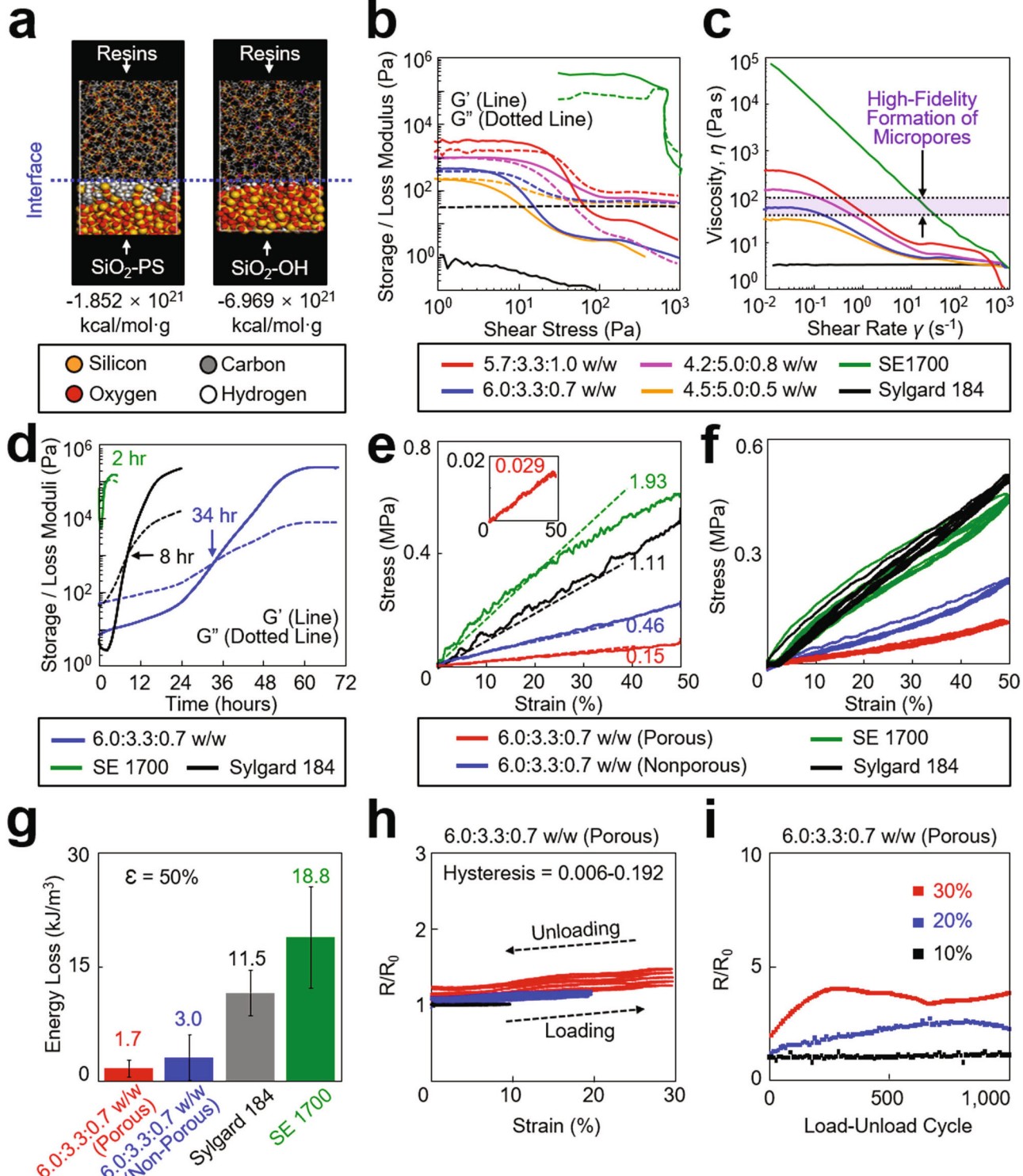

**Fig. 2 Material structure-property relationships. a** Snapshot image of the MD simulation for the interfacial interaction energy of a $SiO_2$-PS particle (left panel) and a $SiO_2$–OH particle (right panel) with the surrounding resins. **b** Storage and loss modulus of the inks with respect to shear stress. **c** Viscosity of the inks with respect to shear rate. **d** Change in the storage and loss modulus of the inks over time. **e** Stress-strain curves of the inks. The inset graph shows the corresponding results obtained from the ink with 6.0:3.3:0.7 ratio configured into an open mesh layout. **f** Mechanical hysteresis of the inks for five loading-unloading cycles at a strain of 50%. **g** Energy loss of the inks ($n = 5$ per group). **h** Electrical hysteresis of the inks with respect to the strain from 10 to 50%. **i** Change in the resistance of the inks throughout 1000 times loading-unloading cycles with the strains ranging from 10 to 30%.

a bare PDMS ink (Sylgard 184, Dow Corning). Figure 2b presents the measurement results for the storage (G′; lines) and loss (G″; dotted lines) modulus of these blends. The inks with 5.7:3.3:1.0, 6.0:3.3:0.7, and 4.2:5.0:0.8 ratios and the SE 1700 ink exhibited a gel-like viscoelastic behavior (i.e., G′ > G″) within the linear viscoelastic (LVE) region (plateau regions) that indicates a printable range without distorting the structural integrity. For example, Supplementary Figure 3 provides a set of optical images for various as-printed (prepolymer) structures with the minimum line width and spacing of 200 μm and 300 μm, respectively. On the other hand, the ink with 4.5:5.0:0.5 ratio and the Sylgard 184 ink exhibited a liquid-like viscoelastic behavior due to their dominating G″ at all shear stresses, which thereby precludes their use for dispensing through a nozzle. The inks exhibited the shear-thinning behavior in which their viscosity decreased with an increase of the shear rate, whereas the Sylgard 184 ink exhibited a zero-shear-rate behavior (plateau curve) at all shear rates (Fig. 2c). The purple-highlighted area in the graph indicates a range of viscosity where the high-fidelity formation of micropores (i.e., sponge-like foam) occurred under the hot pressurized steam condition. The results also show that the ink with 6.0:3.3:0.7 ratio remained within this range at low shear rates. As shown in Fig. 2d, this ink also substantially extended the working lifetime (i.e., the time to reach the crossover between G′ and G″) up to 34 h, as compared to the SE 1700 ink (2 h) and the Sylgard 184 ink (8 hours). This substantially prolonged working lifetime is essential not only to maintain the structural integrity of the as-printed (i.e., prepolymer) ink prior to complete polymerization,[31] but also to provide sufficient time for the micropores to form across the entire thickness of the structure up to 150 μm (Supplementary Fig. 4). For comparison, Supplementary Fig. 5 presents experimental results showing that the formulated ink was completely polymerized after 15 min of annealing at 160 °C (i.e., the plateau curve of G′), whereas the ink mixed with conductive nanofillers (i.e., Ag flakes) remained unpolymerized even after 1 h of annealing at the same condition. In conclusion, the ink with 6.0:3.3:0.7 ratio meets all requirements of both the appropriate shear-thinning flow behavior and the prolonged working lifetime, allowing for rapid custom prototyping of the sponge-like poroelastic foam.

Figure 2e shows the stress-strain curve for a printed line (2 cm-long × 2 mm-wide × 150 μm-thick) made of the ink with 6.0:3.3:0.7 ratio before (blue line) and after (red line) the formation of micropores, compared to control groups made of the SE 1700 ink (green line) and the Sylgard 184 ink (black line). The mechanical modulus of the printed line decreased more than threefold ($E = 0.15 \pm 0.02$ MPa) in the presence of micropores due to their large volumetric porosity (~70%), which remained substantially lower than the control groups (>1.11 MPa) by nearly or more than tenfold. The inset graph presents the corresponding results obtained from this ink configured into an open mesh layout (8 mm × 20 mm), providing an ultralow effective mechanical modulus ($E = 29 \pm 12$ kPa) that is comparable to that of human cardiac tissues ($E = 29–41$ kPa).[33] These results imply that the printed device is capable of gently interfacing across the epicardial surface in a way that imposes minimal stress on the tissue. Figure 2f, g summarizes the mechanical hysteresis and the corresponding energy loss of the printed device under repetitive loading-unloading cycles (>5 times each) at a strain ($\varepsilon$) of 50%, respectively. The printed device exhibited substantially suppressed mechanical hysteresis with the lowest energy loss of $4.3 \pm 0.5$ kJ m$^{-4}$; a value that was substantially lower than the control group made of the SE 1700 ink ($23.6 \pm 8.7$ kJ m$^{-4}$) by more than fivefold. This energy loss gap increased proportionally to the strain (Supplementary Fig. 6a). Figure 2h presents the continuous electrical measurement of

the printed device under stretching up to 30% that corresponds to the maximum strain of the human heart.[34] The results show that the electrical hysteresis of the printed device remained substantially low between 0.006 and 0.192 which remains at least tenfold lower than similar counterparts reported previously.[7,35] The printed device was stretched up to nearly 150% prior to its fracture while maintaining its resistance ($R/R_0$) below 9.0 (Supplementary Fig. 6b). Figure 2i confirms that the resistance remained constant below 5.0 even after more than 1000 cycles of stretching up to 30%. Throughout these tests, no evidence of the delamination or leakage of the embedded conductive nanofillers (i.e., Ag flakes) was observed. Moreover, the sheet resistance of the printed device remained nearly unchanged within a range of variation (0.5–2.5 Ω sq$^{-1}$) when soaked in a bath of distilled (DI) water, phosphate-buffered saline (PBS), and ethanol for 12 h (Supplementary Fig. 7).

**Rapid custom prototyping.** The well-regulated rheological and mechanical properties of the formulated ink drove the exploration for the rapid prototyping of a custom-designed poroelastic biosensor in a few hours per batch of a dozen, which could be also useful for spatiotemporal ECG mapping on various animal models with a wide range of heart sizes and shapes. Figure 3a displays examples of custom-printed devices, each of which was customized to fit with the enucleated piglet, ovine, porcine, and bovine hearts from upper left to bottom right clockwise. Figure 3b schematically illustrates the cross-sectional view of the hearts to compare their sizes. All these devices were able to accommodate the different sizes and shapes of the hearts at various length scales, while simultaneously forming a seamless contact to the irregular epicardial surfaces due to their substantially low bending stiffness ($<8.0 \times 10^7$ GPa μm$^4$). Here, the spatial resolution of these devices (i.e., the number of electrodes within a given region) was determined by the feature resolution of the nozzle injection system (i.e., the minimum nozzle size of about 100 μm). For instance, a total of 64 electrodes (i.e., 32 recording channels) on the device were uniformly distributed across the entire surface of the enucleated porcine heart that is similar in size to the human heart.[36] In this study, a bipolar recording configuration was used not only to reduce common-mode noises such as power line interference but also to suppress the crosstalk for high fidelity recording of ECG signals.[37]

The poroelastic nature and the open mesh layout of these devices also ensured minimal normal (peeling) stress to the epicardium tissue (i.e., the minimum adhesion energy per unit area ≈ 0.5 mJ m$^{-2}$) and thereby induced a strong capillary adhesion at the interface.[28] This aspect allowed the devices to reliably adhere onto the epicardial surface without slipping and be also detached without significant mechanical impact on the epicardium tissue. Supplementary Fig. 8 presents representative epicardial ECG signals obtained from the enucleated porcine heart by applying an artificial ECG waveform (amplitude = 2 mV cm$^{-1}$; frequency = 1 Hz) using a signal generator (Keithley 3390). The corresponding spatiotemporal ECG mapping results are shown in Supplementary Fig. 9 and Supplementary Movie 1. The ECG signals were followed consistently by those generated from the signal generator, confirming that all 64 electrodes seamlessly interfaced with the epicardial surface. No visible electromechanical movement of the enucleated cardiac tissues was observed throughout these measurements. These observations were consistent when the device was integrated with other organs, such as the enucleated porcine liver (Supplementary Fig. 10a). Notably, the quality of the conformal coverage was maintained even when the device was interfaced with the deeply wrinkled surface of a human brain model (Supplementary Fig. 10b).

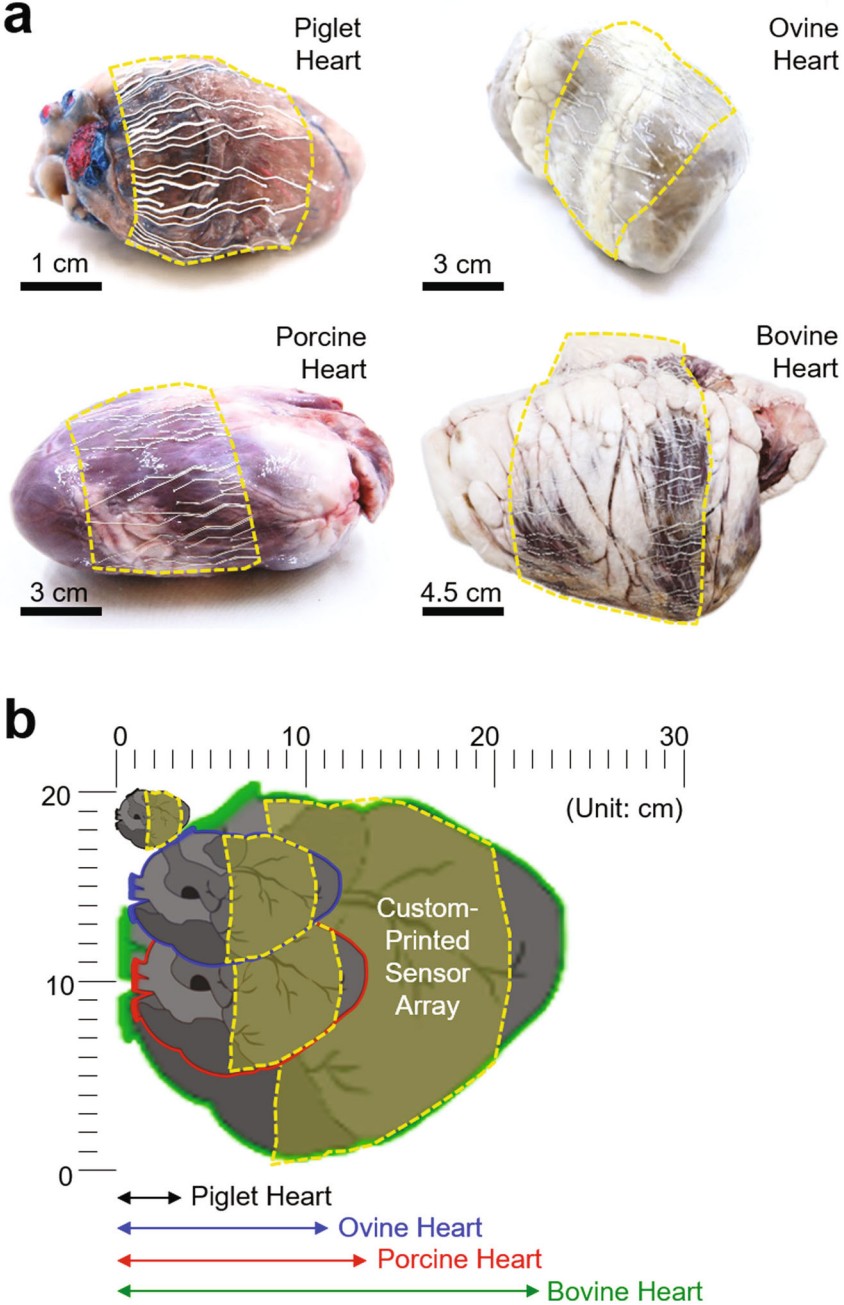

**Fig. 3 Rapid custom prototyping. a** Photographs of the custom-printed sensor arrays, each of which was customized to fit with the enucleated piglet, ovine, porcine, and bovine hearts from upper left to bottom right clockwise. **b** Schematic illustration of the cross-sectional view of the hearts.

**In vivo spatiotemporal recording of epicardial ECG.** We evaluated the ability of the custom-printed devices in the spatiotemporal recording of epicardial electrocardiogram (ECG) in healthy murine ($n = 5$) and porcine hearts ($n = 2$) in vivo. The porcine heart provides a substantial similarity in size and shape to the human heart.[36] The devices were placed on the left ventricle after median sternotomy using a water-soluble film (i.e., PVA). Following the dissolution of the film with the application of a warm saline solution, the devices (~50 μm-thick) adhered intimately to the epicardial surface by capillary adhesion force (Fig. 4a). Here, the devices were configured with a total of 4 and 6 pairs of recording electrodes to cover the total areas of 1.25 cm² and 50 cm² for the acquisition of murine and porcine ECG signals, respectively. The electrochemical impedance of the individual electrodes (200 μm × 200 μm) was characterized in a PBS

solution with a pH of 7.2 at 23 °C as 2.1, 1.5, and 1.0 kΩ at frequencies of 40, 150, and 1000 Hz, respectively (Supplementary Fig. 11). The remarkably low impedances were attributed to the poroelastic properties of the devices that provide not only large interfacial areas between electrodes and electrolytes but also rapid solid-state diffusion of charge carriers.[38] The electrochemical impedance of the as-printed devices was verified prior to their implementation onto the heart in vivo.

The soft and thin nature of the devices enabled them to maintain a highly conformal contact to the epicardial surface under normal cardiac cycles with the murine and porcine heart rates of 529.9 ± 9.3 and 85.4 ± 8.5 beats per minute, respectively. Supplementary Movie 2 shows representative devices that move synchronously with the underlying epicardium tissue without impeding diastolic function in the murine (left panel) and porcine

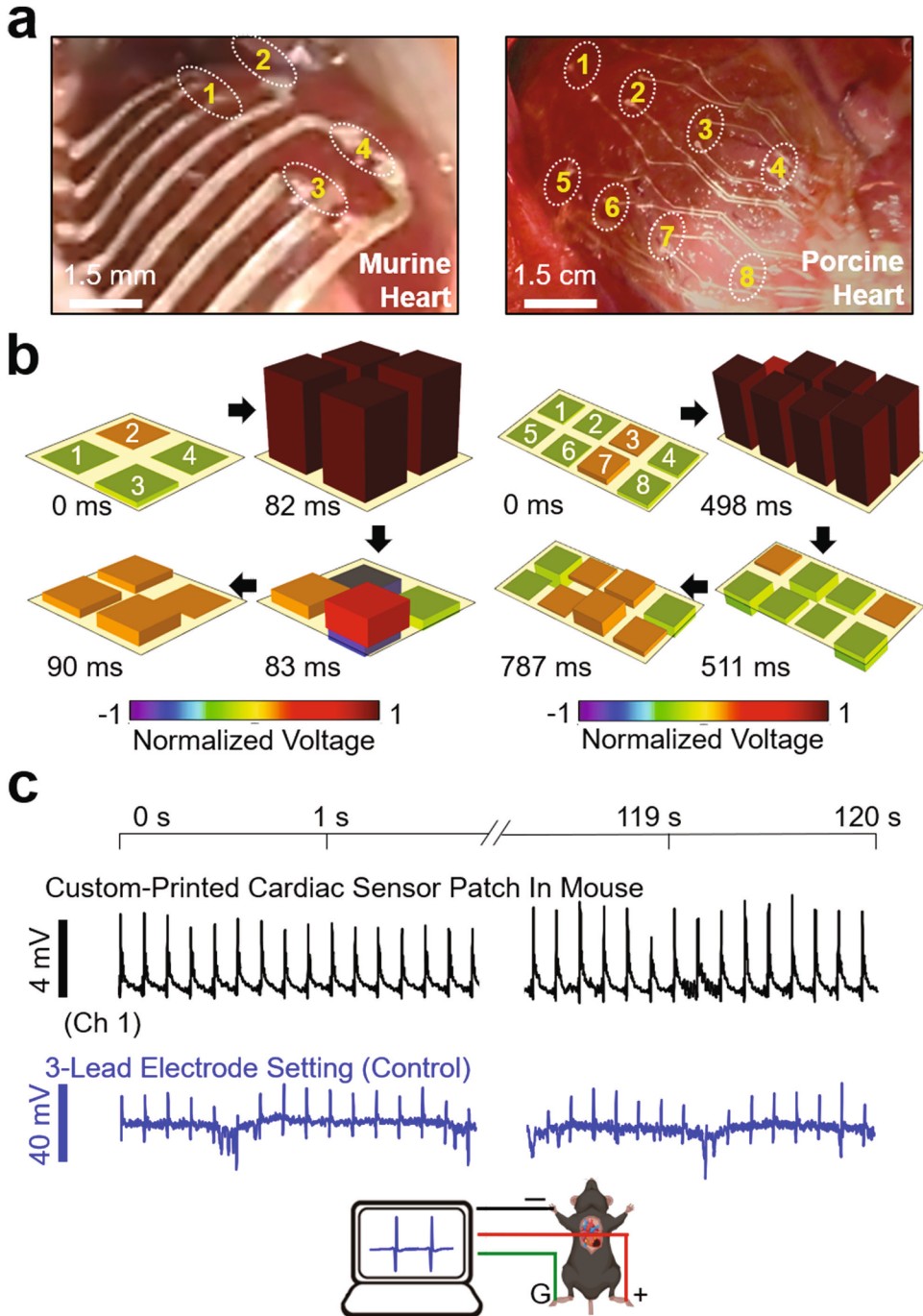

**Fig. 4 Spatiotemporal recording of epicardial ECG signals in vivo. a** Photographs of the custom-printed sensor arrays placed on the epicardial surface of a murine (left panel) and porcine heart (right panel). **b** Post-processed 3D data of the spatiotemporally recorded murine (left panel) and porcine (right panel) ECG signals. **c** Simultaneously measured ECG signals using the custom-printed sensor array (top panel) and a control three-lead electrode set (bottom panel) on a murine heart.

(right panel) hearts. The degree of conformal coverage across the epicardial surface increased with the decreased thickness of the device from 300 μm to 50 μm due to a significant reduction in bending stiffness by more than 200-fold (Supplementary Fig. 12). Figure 4b shows the corresponding measurement results for the post-processed 3D data of spatiotemporally recorded murine (left panel) and porcine (right panel) ECG signals. The raw data of the epicardial ECG signals obtained from all healthy murine and porcine hearts are summarized in Supplementary Fig. 13a, displaying a typical ECG tracing of the cardiac cycle that consists of a P-wave (atrial depolarization), a QRS-complex (ventricular

depolarization), and a T-wave (ventricular repolarization). The corresponding quantitative data of R–R interval, QRS duration, and J-point elevation were measured as $116.3 \pm 3.8/764.3 \pm 65.4$ ms, $2.8 \pm 0.5/28.8 \pm 16.1$ ms, and $-0.1 \pm 0.9/-0.1 \pm 0.4$ mV for the murine/porcine hearts, respectively (Supplementary Fig. 13b). No J-segment elevation was observed in the ECG recordings for the healthy hearts. The strain-insensitive poroelastic behavior (i.e., the negligible electrical hysteresis under cyclic loading with a strain of <30%) of the devices and their robust conformal contact across the beating epicardial surface enabled the high-fidelity acquisition of epicardial ECG signals without noticeable

degradation in signal quality over time (Fig. 4c, top panel). To confirm the change of ECG waveforms over time, Fig. 4c (bottom panel) presents the results of control ECG recording that simultaneously occurred across the body of the mouse (i.e., global ECG signals) using commercial three-lead electrodes (ERT Control/Gating Module Model 1030, SA Instruments, Stony Brook, NY). The amplitude of the global ECG signals was at least threefold higher than those of the epicardial ECG signals while the corresponding R–R interval, QRS duration, and J-point elevation were measured as $118.2 \pm 5.7$ ms, $10.0 \pm 0.8$ ms, and $2.0 \pm 4.5$ mV, respectively. Unlike the epicardium ECG signals obtained from the custom-printed sensor array, the global ECG signals obtained from the control measurement setup clearly displayed the shift (i.e., elevation and depression) of the signal baseline caused by the inhalation and exhalation of breathing, respectively.[37]

**Intraoperative epicardial mapping of infarcted mouse heart**. Intraoperative epicardial mapping is useful in localizing critical regions that indicate the origin of pathophysiological conditions such as arrhythmias after acute myocardial infarction, thereby providing important information to guide definitive surgical treatments, especially when the infarct border needs to be identified.[39,40] To demonstrate the utility of the custom-printed devices in this surgical setting, we performed intraoperative spatiotemporal mapping of epicardial ECG signals in a murine acute myocardial infarction model in vivo. Adult mice underwent left thoracotomy to expose the ventral portion of the heart, followed by the placement of a custom-printed sensor array (a total of six bipolar recording channels) on the epicardial surface to cover the ventricular epicardium (Fig. 5a). Surgery to permanently ligate the left coronary artery was performed following the same procedures as reported in previous studies.[24] Representative results of the epicardial ECG measurements are shown in Fig. 5b. Following approximately 30 s of ligation, ST-segment elevation (red circles in the middle panel) occurred near the ligation point where the sensor channels 2 and 3 were located. After 60 s of ligation, the ST-segment elevation was also detected by the sensor channels 4–6, implying that the regional myocardial infarction propagated toward the apex of the heart with a velocity of approximately $0.6$ mm s$^{-1}$. The ECG signals displayed a convex ST-segment, indicating that the corresponding regions experienced ischemia or hypoxic conditions.[41] A total of nine infarction ECG data (i.e., ST-elevation) captured from three different sensor arrays on mice ($n = 5$), i.e., $1 \times 1$ ($n = 3$), $2 \times 2$ ($n = 1$), and $2 \times 3$ ($n = 1$), showed consistent results as summarized in Supplementary Fig. 14a, along with the corresponding quantitative data of R–R interval, QRS duration, and J-point elevation in Supplementary Fig. 14b. The data exhibited the prolongation of QRS duration, the elevation of J-segment, the metrics of systolic dysfunction, and the elevation of ST-segment after about 40 s of ligation, showing statistical differences from those obtained before ligation using one-way ANOVA with Tukey's post hoc test with significance is set at $p < 0.05$. Figure 5c presents the results of control measurements obtained simultaneously using a three-lead electrode set, displaying reciprocal ST-segment depression to confirm the occurrence of an ischemic event. The control measurements also displayed both reciprocal ST-segment depression and elevation within seconds of left coronary artery ligation (Supplementary Fig. 15), which typically occurs in the three-lead electrode recording configuration.[42–45] This ST-segment depression and elevation were consistently observed during the surgery and throughout the recording duration of 30 min after ligation (Supplementary Fig. 16). The results also imply that the devices maintained a robust and intimate coupling to the epicardial

surface without changing position throughout the recording period (30 min) that involved more than 10,000 individual beats.

The semi-transparency of the devices, due to the open mesh layout and thin-film design (50 μm-thick), enabled simultaneous ultrasound mapping, as a means for validating the location and size of the myocardial infarction region in real-time.[46] Figure 5d schematically illustrates an experimental setup that includes a high-frequency ultrasound system (Vevo3100, FUJIFILM Visual-Sonics Inc), with the enlarged photograph in Supplementary Fig. 17. A warmed ultrasound gel (Ultrasound Transmission Gel, Parker, Inc) was applied directly to the device placed on the left ventricle of the heart. Figure 5e shows a representative short-axis ultrasound image, clearly visualizing the device (yellow circle) along with the epicardium (blue circle) and endocardium (red circle) of the heart. The corresponding real-time ultrasound video is shown in Supplementary Movie 3. The results show that QRS-complex appeared at the initiation of the left ventricle contraction while P-wave prior to the QRS-complex corresponded to the atrial kick responsible for pushing residual blood from the left atrium to the left ventricle (i.e., the phase of diastolic left ventricular filling). The overall quality of both the ultrasound images and the simultaneously recorded ECG signals were unaffected by motion artifacts (e.g., the heartbeat and respiration) through the robust conformal coupling to the epicardial surface. For comparison, Supplementary Fig. 18 provides the ultrasound images of both relatively thick (200 μm-thick; top panel) and thin (50 μm-thick; bottom panel) devices placed on the epicardial surface of a fixed murine heart. Supplementary Movie 4 shows the corresponding real-time ultrasound videos during the removal of the devices from the epicardial surface, clearly visualizing the movement of shadows. These results confirm that the imaging artifacts (e.g., the shadow of the recording electrode pairs) were minimized with the decreased thickness of the device. Figure 5f presents a post-processed 3D image reconstructed from both the ultrasound images and the spatiotemporally recorded ECG signals after 60 s of ligation. The corresponding real-time display of the 3D images is shown in Supplementary Movie 5, confirming that the acute ST-segment elevation was detected by the sensor channels 2–6 (i.e., positive voltages) while the areas nearby the sensor channel 1 remained unaffected (i.e., negative voltages).

**Biocompatibility, anti-biofouling, cardiac function assessment**. Evaluation of the in vivo biocompatibility and anti-biofouling properties of the custom-printed devices and their effect on cardiac function is a critical factor.[47] To this end, we first evaluated the cellular toxicity and inflammatory response of a printed device. Figure 6a shows the result of a cell compatibility assay for the device that was seeded with heart myoblast (H9C2) cells in a 24-well plate (Fisher Scientific, USA), as measured using a colorimetric assay kit (MTT 3-(4,5-dimethylthiazol-2-yl)-2,5-diphenyltetrazolium bromide, Sigma-Aldrich, USA). The results indicate that the proliferation rate of the cells increased consistently throughout the assay period (24 h), producing no significant difference compared to a control (black bar) and a bare sponge-like foam (green bar). Whereas, the device without the overcoat of Au (red bar) showed considerably reduced cell variability (<70%), suggesting toxicity. Figure 6b shows representative histological cross-sectional views of the murine cardiac tissues that were stained with both hematoxylin-eosin (H&E; left panel) and Masson's trichrome (MTC; right panel) on day 14 post-implant of the device. The results revealed moderate chronic inflammation including the formation of a granuloma for the 14-days implantation. Supplementary Fig. 19 provides an overview of nearby granuloma and aorta on day 7 post-implant of the device. The magnified views of granuloma, macrophages, and

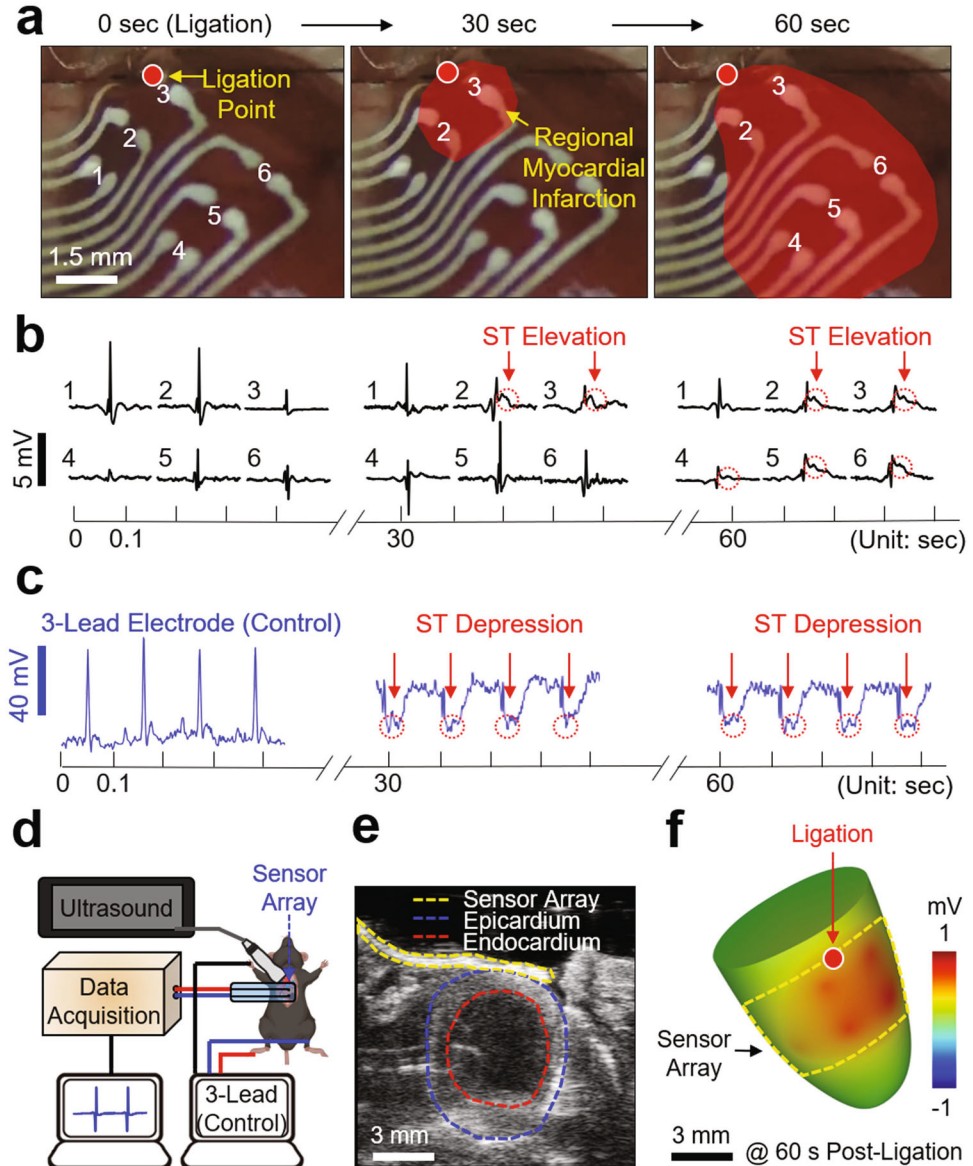

**Fig. 5 Intraoperative epicardial mapping in a murine myocardial infarction model. a** Enlarged images of the custom-printed sensor array covering the entire ventricular epicardium. The red highlighted area indicates the regional myocardial infarction propagating toward the apex of the murine heart over time. **b** Measured epicardial ECG signals using the custom-printed sensor array. **c** Simultaneously measured ECG signals using a control 3-lead electrode set. **d** Schematic illustration of the experimental setup for simultaneous epicardial ECG recording and ultrasound imaging. **e** Representative short-axis ultrasound image displaying the custom-printed sensor array (yellow circle) along with the epicardium (blue circle) and endocardium (red circle) of the heart. **f** Post-processed 3D image reconstructed from the spatiotemporally recorded epicardial ECG and ultrasound signals after 60 s post-ligation.

multinucleated giant cells at the surface of the implanted device suggest chronic inflammation. Supplementary Fig. 20 presents an increased thickening of epicardium near the implanted device on days 1, 7, and 14 post-implants, showing its progression from $44.4 \pm 8.3\ \mu m$ to $645.9 \pm 5.3\ \mu m$ in thickness. The results indicate that chronic inflammatory response directed towards the implanted device. The corresponding chronic epicarditis on days 7 and 14 post-implants for pathological evaluation are shown in Supplementary Fig. 21. Detailed discussions of the histological analysis are also summarized in "Methods".

Next, we also evaluated the biofouling resistance of the device by quantifying the surface fluorescence intensity after 2 h of incubation in $6\ mg\ ml^{-1}$ of a bovine serum albumin-fluorescein conjugate (BSA-FITC; A23015, Fisher Scientific, USA) diluted with 1× PBS, as compared to control groups made of the SE 1700 ink and the Sylgard 184 ink and prepared on a pristine glass

(Fig. 6c). Figure 6d presents the corresponding results of one-way repeated measures analysis of variance (ANOVA) tests ($n = 5$ per group), showing statistical differences between groups. The results show that the fluorescence intensity of the device ($0.7 \pm 0.5$ a.u.) remained significantly lower than that obtained using the Sylgard 184 ink ($10.2 \pm 5.5$ a.u.) and the glass ($15.4 \pm 6.1$ a.u.), suggesting that the porous surface of the device effectively prevented the accumulation of proteins.[48,49] The control group made of the SE 1700 ink produced a statistically comparable degree of biofouling resistance ($2.8 \pm 1.7$ a.u.).

Finally, we evaluated the effect of the device on cardiac function when implanted on the epicardial surface of the murine heart. On days 0, 1, 7, and 14 post-implants, we acquired ultrasound images of the left ventricle in a long-axis (LAX) plane (Fig. 6e and Supplementary Movie 6). With these LAX ultrasound images, the endocardial surface of the left ventricle was

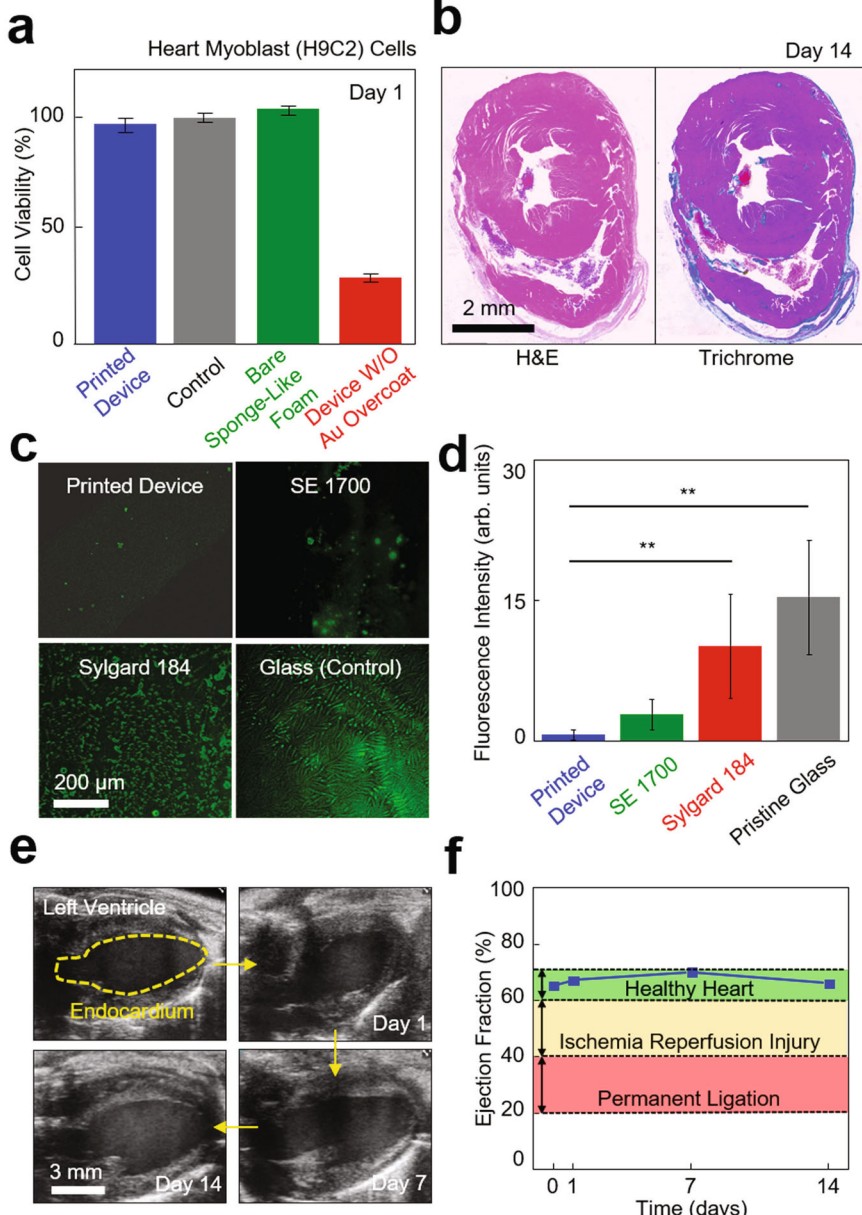

**Fig. 6 Evaluation of biocompatibility, anti-biofouling, and effect on cardiac function. a** Results of cell compatibility assay for the custom-printed sensor array with heart myoblast (H9C2) cells in a 24-well plate, as measured using a colorimetric assay kit ($n = 3$ per group). **b** Representative histological cross-sectional views of the murine cardiac tissues that were stained with both hematoxylin-eosin (H&E; left panel) and Masson's trichrome (MTC; right panel) on day 14 post-implant of the custom-printed sensor array. **c** Z-stack fluorescence images of a BSA-FITC (6 mg/ml) coated on the printed device as compared to control groups made of the SE 1700 ink and the Sylgard 184 ink and prepared on a pristine glass, from upper left to bottom right clockwise. **d** Results of one-way ANOVA tests with Bonferroni correction ($n = 5$ per group). Fluorescence intensity of pristine glass and sylgard 184 is shown as average ± standard deviation (**$p = 3.4 \times 10^{-7}$, $4.1 \times 10^{-4}$), versus printed device. **e** Representative ultrasound images of the left ventricle of a murine heart on 0, 1, 7, and 14 days implantation of the custom-printed sensor array. **f** Results of the ejection fraction of the murine heart post-implant.

segmented at both end-diastole and peak-systole and used to calculate end-diastolic volume (EDV) and peak-systolic volume (PSV) using the Simpson's rule of disks.[50] The ejection fraction (EF) (i.e., the percentage of blood pumped by the left ventricle during contraction) was then calculated using these volumes to assess global cardiac function.[51] The results confirm that the EF of the heart remained within the normal range (60–70%; green highlighted area) without noticeable decrease throughout the entire implantation period (Fig. 6f), which was clearly higher than the abnormal ranges for ischemia-reperfusion injury (40–60%; yellow highlighted area) and permanent ligation (20–40%; red highlighted area).

## Discussion

The results reported herein suggest a route towards rapid prototyping of thin and stretchable poroelastic biosensors with a custom-fit design that can meet a specific geometric demand in clinical practices. The determination of a new formula for a dispensable silicone ink leads to optimal rheological properties that enable both (1) the high-precision DIW of arbitrary functional microarchitectures at various length scales and (2) the capability of turning the printed microarchitectures into a sponge-like foam in a deterministic manner. Uniquely, the resulting devices are monolithic in which the densely networked conductive nanofillers are embedded through the internal pores

in a way that eliminates the risk of compromising their structural integrity even under large deformations. The poroelastic nature of the devices can be used to establish a robust coupling to the epicardial surface and also remain insensitive to periodic cardiac cycle and respiratory motion without significant mechanical and electrical hysteresis. The simultaneous intraoperative monitoring of both epicardial ECG and ultrasound signals in a murine acute myocardial infarction model suggests a potential utility of the device for high-fidelity acquisition of real-time 3D cardiac mapping, which may guide surgical interventions such as ablation for arrythmias. The in vivo studies also suggest an opportunity to further increase the spatial resolution of these devices (i.e., the number of electrodes within a given region) in order to alleviate the need for reliance on post-processing algorithms to map the site of myocardial infarction in a higher resolution. The basic concept of this approach may be expandable for the continuous monitoring of lethal cardiac diseases through chronic implantation of the devices and integration with current state-of-the-art means of wirelessly communicating power and data.[52]

## Methods

**4D heart segmentation.** All ultrasound images were acquired using the Vevo 3100 small animal ultrasound system (FUJIFILM VisualSonics Inc., Toronto, Canada). The 4D ultrasound data (3D geometric volume over a cardiac cycle) of adult mouse hearts with well-developed infarcts were acquired via high-frequency ultrasound. The cardiac and respiratory-gated 2D short-axis images were acquired from the apex to the base of the left ventricle and spatiotemporally synced to generate the 4D ultrasound data. The reconstructed data was resampled to isotropic voxels and exported to SimVascular for the 3D segmentation at both end-diastolic and peak-systolic timepoints.[25] STL files of the myocardial wall were then created with uniform meshing and used to design the devices.

**Ink composition and preparation.** The formulated inks were prepared by blending the following three compositions: a base resin, Sylgard 184, and $SiO_2$-PS silica particle, in a specific weight ratio (5.7:3.3:1.0, 4.2:5.0:0.8, 6.0:3.3:0.7, and 4.5:5.0:0.5) with a mixer (Thinky Mixer) for 5 min at 2000 rpm. The base resin was prepared by blending 64.5 wt% of vinyl terminated diphenylsiloxane-dimethylsiloxane copolymer (a mixture of 9:1 weight ratio of PDV-0541 and PDV-0525), 33.5 wt% of trimethylsiloxane terminated methylhydrosiloxane-dimethylsiloxane copolymer (HMS-151), and 2.0 wt% of 1,3,5,7-tetravinyl-1,3,5,7-tetramethylcyclotetrasiloxane (siloxane monomer). Sylgard 184 (a mixture of 10:1 weight ratio of base and curing agent) was used as a dilute resin. The $SiO_2$-PS particles and Pt-carbonyl cyclovinylmethylsiloxane complex (Pt catalyst; 0.1 wt% of the amount of base resin) were added and blended using a mixer (Thinky Mixer) for 2 min at 2000 rpm to adjust the rheological properties of the ink. The control inks (e.g., Sylgard 184 and SE 1700) were prepared by mixing the base and curing agent at the standard 10:1 weight ratio.

**Mechanical and electrical measurements under cyclic strains.** The well-mixed blends for the formulated inks with a certain ratio were spin-casted on a water-soluble film (PVA, Sigma-Aldrich, 10 μm-thick) to form a thin layer of thickness of 150 μm. Any air bubbles introduced during the casting process were removed before placing the lids on the molds. The following polymerization at 120 °C for 30 min in an oven, the specimens were allowed to cool down to room temperature for about 15 min and then trimmed into a rectangular shape with a width of 2 mm. Tensile strength was measured using a mechanical testing system (Mark-10). The specimens were loaded into uniaxial grips and then pulled to reach a breakpoint at a speed of 20 mm min$^{-1}$. A total of five trials per specimen was taken, and the corresponding standard deviation was reported as a measurement error. For the electrical measurement, a source meter (Keithley 2400, Tektronix) and a custom-built LabView code (National Instruments) was used in a two-wire configuration.

**Measurement of rheological properties.** Rheological measurements were carried out on a Rheometer (Discovery Hybrid Rheometer-2, TA Instrument) using parallel plate fixtures with a diameter of 25 mm. The viscosity and shear moduli were measured by stress sweeps ranging from 0.1 to 65,000 Pa at a fixed angular frequency of 10 rad s$^{-1}$ for each measurement. Oscillatory time sweeps were measured to investigate gel-points (i.e., working lifetime), where the storage modulus (G′) becomes larger than the loss modulus (G″) at an angular frequency of 10 rad s$^{-1}$ and a strain rate of 4%.

**Molecular dynamics (MD) simulation.** The Forcite and Amorphous Cell modules in Material Studio (BIOVIA, UK) were used for the MD simulation. The density of the surface-treated layer, composed of one vinyl terminated diphenylsiloxane-dimethylsiloxane copolymer and three trimethylsiloxane terminated methylhydrosiloxane-dimethylsiloxane copolymer main-chains, was set to 0.97 g cm$^{-3}$. Geometry-optimized 3D models were followed by the anneal protocol in which the temperature of the system was sequentially set to 298, 398, 498, and 598 K, and then decreased in reverse using a constant-volume ensemble (NVT). Each step was performed for 50 ps, and the annealing protocol was repeated five times. The interaction energies of the 3D models were obtained after additional NVT dynamics at 298 K for 1000 ps. The condensed-phase optimized molecular potentials for atomistic simulation studies (COMPASS) II force field, the Ewald summation method for non-bonding interactions with an accuracy of 0.001 kcal mol$^{-1}$, the time step of 1.0 fs, the Andersen temperature control method with 1 as the collision ratio, and the Berendsen pressure control method with 0.1 ps as the decay constant were chosen.

**Fabrication of custom-printed sensor array.** The process began with a Si wafer coated with a thin layer (1 μm-thick) of polymethyl methacrylate (PMMA) as a chemically dissolvable sacrificial layer. The surface of the PMMA layer was exposed with 3-aminopropyltriethoxysilane (APTES) in a vacuum desiccator to form a hydrophilic silane group to improve the adhesion strength. A direct writing of the formulated ink was followed using the nozzle injection system (Nordson EFD). For the formation of micropores, ink was annealed in a pressure rice cooker (Max, Instant Pot, Inc.) in which the steam temperature and pressure were set at 120 °C and 15 psi with the ramping rate of 15 °C per min and 5.6 psi per min, respectively. Next, the resulting structure was immersed in a mixture solution (200 ml) of hexane and Ag flakes (200 nm–5 μm in diameter; inframat advanced materials, LLC) to trap the Ag flakes into the internal pores. The structure was then immersed in a Cu plating solution that contains (1) copper(II) sulfate pentahydrate (CuSO$_4$·5H$_2$O, Sigma-Aldrich; 18 g L$^{-1}$), (2) ethylenediaminetetraacetic acid (EDTA, Sigma-Aldrich; 48 g L$^{-1}$), (3) potassium hexacyanoferrate(II) trihydrate (K$_4$[Fe(CN)$_6$]·3H$_2$O, Sigma-Aldrich; 600 mg/L), (4) sodium hydroxide (NaOH, Fisher scientific; 45 g L$^{-1}$), (5) poly(ethylene glycol) (H(OCH$_2$CH$_2$)$_n$OH, Sigma-Aldrich; 500 mg L$^{-1}$), (6) formaldehyde (HCHO, Fisher scientific; 20 mL L$^{-1}$), and (7) hydrochloric acid solution (1 N) (HCl, Fisher Scientific; 18 mL L$^{-1}$) for 30 min. The Cu-plated surface was subsequently plated with Au for 2 min using an electroplating system (24 K Pure gold plating solution-Bath, Gold Plating Services), followed by thorough rinsing with deionized (DI) water. A layer (~50 μm-thick) of polyvinyl alcohol (PVA, Sigma-Aldrich) was drop-casted to form the temporary handling support. Following the complete dissolution of the bottom PMMA layer with acetone, the complete device was physically separated from the Si wafer by gently peeling the PVA layer. Trimming the excess area of the PVA layer completed the entire process.

**Animal surgeries on mice.** All surgical procedures on mice have performed aseptically and approved by the Purdue Animal Care and Use Committee under protocol number 1505001246. Adult male mice (> 12 weeks old; wild-type; C57BL/6J; The Jackson Laboratory, Bar Harbor, ME) were used for this study. Mice were group-housed (up to five mice in a single cage) in an 18–23 °C room with 40–60% humidity under a 12-h light/12-h dark cycle. Mice were given ad libitum access to food (standard chow) and water with Nestlets as nesting material. To prepare for surgery, each mouse was anesthetized with 1–3% isoflurane delivered in 100% O$_2$ and endotracheally intubated to a small animal ventilator (SomnoSuite, Kent Scientific, Torrington, CT). To prevent pneumothorax, pressure-controlled ventilation was employed to maintain a target inspiratory pressure of 18 cm H$_2$O and a peak-end expiratory pressure of 5 cm H$_2$O in the lungs. The mouse was secured to a heated surgical stage, and the body temperature was kept between 36 °C and 37 °C using a homeothermic control module. A three-lead needle electrode set (ERT Control/Gating Module Model 1030, SA Instruments, Stony Brook, NY) was positioned in a Lead I configuration to continuously collect ECG waveforms throughout the surgical procedure at a sampling rate of 900 Hz (Matlab, MathWorks Inc.). A small incision was made along with the third intercostal space of the left thorax, and the ribs were retracted to expose the left ventricle before dissecting the pericardium to identify the left coronary artery. For acute infarction studies (n = 5), an 8–0 nylon suture was loosely looped around the left coronary artery, and the testbed custom-printed devices were placed on the epicardial surface using a water-soluble film (PVA; 50 μm-thick, Sigma-Aldrich, USA). Warmed sterile saline was applied to completely dissolve the water-soluble film within no more than 30 s. A suture was tightened to permanently ligate the left coronary artery in order to induce an acute infarct. Successful ligation was confirmed by discoloration of the myocardium in regions distal to the ligation site and global ST-segment elevation or depression in the three-lead electrode set. The ECG data from both the printed devices and the control three-lead electrodes were acquired simultaneously and synchronized by their timestamps. The mice in the acute infarction group were euthanized humanely at the end of the procedure. For implantation surgeries (n = 3), we placed the printed device (50 μm-thick) to the epicardial surface as mentioned previously, sutured close the incision site, and allowed the mice to recover. Buprenorphine (0.05 mg kg$^{-1}$ animal body weight) was administered via intraperitoneal injection as an analgesic. The left ventricles of these mice were imaged with ultrasound on days 1, 7, and 14 post-implants, and one mouse was euthanized at each time point for longitudinal histological data.

**Animal surgeries on pigs**. All surgical procedures on pigs were terminal, performed by a trained veterinary team from Purdue University's College of Veterinary Medicine, and approved by the Purdue Animal Care and Use Committee under protocol number 1406001099. Adult domestic pigs ($n = 2$) were anesthetized and intubated with a ventilator throughout the entire procedure. A median sternotomy was conducted and the pericardium was cut to visualize the anterior wall of the left ventricle. The custom-printed devices, configured with a total of 16 electrodes i.e., eight bipolar recording channels, were then placed onto the epicardial surface of the left ventricle using a water-soluble film. Warmed sterile saline was applied to dissolve the water-soluble film for a tight conformal contact, and ECG waveforms were recorded continuously.

**Simultaneous ultrasound imaging during ECG recording**. All ultrasound images were acquired using the Vevo 3100 small animal ultrasound system (FUJIFILM VisualSonics Inc., Toronto, Canada). For the preparation prior to imaging, mice were anesthetized with 1–3% isoflurane delivered in medical-grade air and positioned supine on a heated imaging stage, with paws secured to gold-plated stage electrodes to monitor for ECG and respiration signals. A 40 MHz central frequency linear array transducer with 256 elements (22–55 MHz; MX550D) was then positioned on the left ventral thorax to acquire ultrasound images in multiple long- and short-axis planes. For the data acquisition, both ECG and respiratory gatings were implemented to minimize breathing artifacts and to acquire cardiac cine loops at 1000 Hz. To evaluate left ventricular function, the endocardial surface of the left ventricle was manually segmented at end-diastole and peak-systole to approximate left ventricular volumes using Simpson's rule of disks.[51] EF was then calculated as follows:

$$EF = \frac{EDV - PSV}{EDV} \times 100 \qquad (1)$$

where EDV and PSV correspond to EDV and PSV, respectively. For the in vivo open chest imaging, a warmed ultrasound gel was applied directly onto the epicardially-implanted device on the beating heart at the end of several infarction surgeries. Several representative short-axis images of the left ventricle were acquired to visualize adherence of the patch to the left ventricle. For the ex vivo, ultrasound imaging on fixed tissues, the devices (50–300 μm-thick) were placed on the epicardial surface of the previously fixed-left ventricles. An ultrasound gel was used as a conductive medium between the transducer and the fixed tissue. Multiple short-axis images of the left ventricles were acquired to investigate how the devices impact the quality of ultrasound imaging.

**Anti-biofouling analysis**. The biofouling resistance of the printed devices was analyzed following the same procedures as reported in a previous study. The specimens were incubated in bovine serum, fluorescein conjugate (BSA-FITC; 6 mg ml$^{-1}$) diluted in 1× phosphate-buffered solution for 2 h in a 6-well plate protected from light at room temperature ($n = 5$ per group). The specimens were then rinsed with 1× phosphate-buffered solution prior to air drying and imaging. The fluorescence z-stack images (Zeiss Zen Black 2.3) were captured at 10× magnification using an inverted fluorescence microscope (Axio Observer Z1, Carl Zeiss Microscopy, Jena, Germany). The fluorescence intensity was quantified using a Java-based image processing program (ImageJ). All images were acquired using the same exposure settings to ensure no over- or under-saturated pixels (pixel values = 0–255) or image histogram aberrations. The mean intensity and standard deviation were measured from whole image selections, and the statistical analysis was performed with the SAS software (SAS Institute).

**Histological analysis**. At the end of the implantation experiment, mice were euthanized humanely. Retrograde perfusion was performed from the inferior vena cava using 30 mM potassium chloride (KCl) solution to clear residual blood and arrest the heart in diastole. The heart was harvested while minimizing any disruption to the implanted device on the epicardial surface and then fixed in 4% paraformaldehyde for 7 days at 4 °C before sending the samples for histology. The hearts were embedded in paraffin, thinly sectioned (5 μm-thick), and stained with hematoxylin-eosin (H&E) and MTC. The stained tissues were imaged in segments at 10×, 20×, and 40× magnification using a Leica ICC50W stereomicroscope (Leica Microsystems Inc., Buffalo Grove, IL), and stitched together with MosaicJ. Epicardial thickening was measured from histological images using ImageJ. Microscopic examination was performed by a board-certified veterinary pathologist and the interpretation was based on standard histopathological morphology of a murine heart. Four serial transverse sections of the heart from the base to the mid-papillary region were histologically evaluated. The results revealed that, on day 7 post-implant, the implanted device was present adjacent to the wall of the aorta invoking the formation of a granuloma (Supplementary Fig. 19). The granuloma was comprised of a central necrotic core of eosinophilic cellular and karyorrhectic debris mixed with numerous degenerate and viable neutrophils. Surrounding the necrotic core were epithelioid macrophages and multinucleated giant cells, rimmed peripherally by lymphocytes, plasma cells, fibroblasts, and fibrous connective tissue. The epicardium of the right and left atrium was expanded by fibroblasts admixed by neutrophils, lymphocytes, and macrophages (Supplementary Fig. 20). The fragments of the foreign materials surrounded by inflammatory cells were found adjacent to the wall of the right atrium. On day 14 post-implant, the epicardium of the right ventricle was thickened with pale eosinophilic collagen fibers and increased fibroblast cellularity.

The pericardium was similarly thickened with collagen, numerous fibroblasts, small-caliber blood vessels, and infiltrates of lymphocytes, plasma cells, macrophages, multinucleate giant cells, and few neutrophils. The thickened pericardium surrounded fragments of granular, black foreign material with multifocal adhesions to the underlying epicardium (Supplementary Fig. 21). The observed lesions predominantly bordered the right ventricular free wall. The pericardium was only observed on the right-side following tissue processing, likely because the pericardium has adhered to the underlying inflamed epicardium. Although chronic inflammation due to a foreign body response was present at the implant site after implantation, its effects on intraoperative epicardial mapping or cardiac EF were insignificant. While short-term intraoperative implantation side effects cannot be ruled out, we observed no issues during intraoperative epicardial mapping of cardiac EF. The inflammation could be further reduced through the inclusion of selective anti-biofouling surface coating (except for the areas of the Au recording electrodes) or the application of nanoscale texturing across the outer surface of the devices.[53–55] The nano-textured surface of the devices could further improve the adhesion to the epicardium owing to the hydration characteristics at the interface.[56]

**Data processing**. The ECG data were collected using a multi-channel data acquisition system (ADInstruments) at a sampling rate of 4 kHz and 1 kHz for the murine and porcine hearts, respectively. The data were then filtered using a 5 Hz high-pass filter via the embedded LabChart software (Supplementary Fig. 22). The data were expressed as a matrix with n (the number of bipolar recording channels) × time based on the spatial location of the recording electrodes and then interpolated using a custom Python code for smooth fadeaway of the colors (i.e., voltages) in the spatiotemporal 3D ECG mapping results.

**Statistics and reproducibility**. Micrograph images presented in Fig. 1b–d and Supplementary Fig. 4 are representative SEM images from one of over 100 successful poroelastic biosensor arrays. The representative short-axis ultrasound image in Figs. 5e and 6e and Supplementary Fig. 18 are from one of over three images. All histological images in Fig. 6b and Supplementary Figs. 19ac, 20ac, and 21 from three experiments.

## Data availability

The data that support the findings of this study are available from the corresponding authors upon reasonable request.

## Code availability

The code that support the findings of this study are available from the corresponding authors upon reasonable request.

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

## Acknowledgements

We thank Bill Schoenlein and Melissa Bible for their help with the porcine surgeries. C.H.L. acknowledges funding supports from the Asian Office of Aerospace Research & Development (AOARD: FA2386-16-1-4105; program manager: Dr. Tony Kim) and the Air Force Office of Scientific Research (AFOSR: FA2386-18-1-40171; program manager: Dr. Tony Kim). C.H.L. also acknowledges funding supports from the SMART film by the Birck Nanotechnology Center at Purdue University. K.-S.L. acknowledges support from the US Department of Energy's National Nuclear Security Administration contract DE-AC52-06NA25396 and the Dynamic Materials Properties Campaign. The Los Alamos National Laboratory is an affirmative action equal opportunity employer, managed by Triad National Security, LLC for the U.S. Department of Energy's NNSA, under contract 89233218CNA000001. H.L. acknowledges funding support from the National Science Foundation (United States) under grants ECCS-1944480 and CNS-1726865. C.H.P. acknowledges funding support from the National Research Foundation of Korea (NRF) Grant funded by the Korea government (MSIT) (NRF-2017R1C1B3009270 and 2019R1A2C1087209). D.R.K. acknowledges funding support from the International Research and Development Program (NRF-2018K1A3A1A32055469) and the Basic Science Research Program (NRF-2018R1C1B6007938) through the National Research Foundation of Korea (NRF) funded by the Ministry of Science and ICT of Korea. D.R.K acknowledges funding support from the International Research and Development Program (NRF-2018K1A3A1A32055469) through the National Research Foundation of Korea (NRF) funded by the Ministry of Science and ICT of Korea. H.J. acknowledges the funding support from the MOTIE (Ministry of Trade, Industry, and Energy) in Korea under the Fostering Global Talents for Innovative Growth Program (P0008748, Global Human Resource Development for Innovative Design in Robot and Engineering) supervised by the Korea Institute for Advancement of Technology (KIAT). Additional funding was provided by the Leslie A. Geddes Endowment at Purdue University (C.H.L. and C.J.G.).

## Author contributions

C.H.L., C.J.G., and K.S.L. conceived the concept, planned the project, and supervised the research. C.H.L., C.J.G., K.S.L., B.K, A.H.S., and W.P. designed and conducted the experiments and data analysis. J.Z., N.S.G., K.K., Y.J., H.J., and D.R.K. characterized the mechanical and electrical properties of the devices. W.P., A.C., H.M., H.L., and A.H.S. evaluated the biocompatibility, biofouling, and effect on cardiac function of the devices. C.H.P. developed the numerical codes and carried out the computational simulations. C.H.L., C.J.G., K.S.L., B.K., and A.H.S. wrote the manuscript. All authors analyzed the data and commented on the paper.

## Competing interests

The authors declare no competing interests.
