## [Peer Review File · Nature Communications]

REVIEWER COMMENTS

Reviewer #1 (Remarks to the Author):

The authors in this manuscript demonstrated a sponge-like form of silicone composite with excellent rheological properties for printing in a nozzle injection system with rapid prototyping capability. This composite material simultaneously integrated the following features: 1. low hysteresis, from its poroelastic behavior instead of viscoelastic behavior with reversible compressibility that can effectively suppress both mechanical and electrical hysteresis against repetitive loading; 2. exceptional softness, due to the ultralow mechanical modulus (<30 kPa) of the sponge-like foam, which is lower than that of commercial dispensable inks by more than one order of magnitude and comparable to that of human cardiac tissues; 3. monolithic structure, since the nanofillers are embedded through the internal pores in a way that eliminates the risk of compromising the structural integrity even under large deformations. The authors further elucidated the comprehensive structure-property relationships of this composite material at the molecular and microsystemic levels and then evaluated its applicability in rapid custom prototyping of stretchable biosensors. Finally, they produced a range of custom-fit biosensor arrays of very different dimensions tailored for simultaneous recording and imaging of acute myocardial infarctions in vivo.

Overall, this manuscript addresses an important challenge in the area of direct ink writing of functional biomedical devices, namely, existing inks exhibit significant mechanical and electrical hysteresis under periodic large strain cycles, compromising their usage in various applications. And to the knowledge of this reviewer, the approach the authors is taking regarding the ink design, formation and fabrication is highly novel. The manuscript also has great depth and breadth regarding the studies of fundamental poroelastic materials science and demonstrations of important cardiac bio-electronic systems. This reviewer therefore highly recommends this manuscript to be published on Nature Communications.

Minor comments:

1. On page 3, the unit for sheet resistance should be Ohm per square instead of just Ohm.
2. Both Ag and Cu are not used often in implantable devices due to their limited biocompatibility (as also demonstrated in their in vitro cell culturing studies), although in this manuscript they are covered by Au. Can the authors comment on the possibility of getting rid of them entirely from the system?
3. A related issue to the above is the adhesion of metal to the silicone porous substrate. From the R/R0 and strain test results and the cyclic stretching test the interface seems to be very stable. Can the authors perhaps elaborate on what leads to the stable interface between Ag and silicone?
4. In addition to stretching test, the other important reliability measurement is soak test. The authors are highly recommended to look at the performance of their DIW electrodes as a function of soaking.
5. The 200 micron by 200 micron electrode from their direct ink writing is actually having remarkably small impedance of only 2.1 kilo Ohm at 40Hz. This is truly high performance, largely due to the porous structure in the electrodes.

Reviewer #2 (Remarks to the Author):

Kim et al. "Rapid Custom Prototyping of Soft Poroelastic Biosensor for Simultaneous Epicardial Recording and Imaging" presented potentially impactful new technology for stretchable biosensors that could be produced using direct ink writing approach. Presented poroelastic materials possess impressive mechanical properties, which allow such devices to conform to soft and mechanically active biological tissues, such as the heart. An additional benefit of these materials is their compatibility with ultrasound imaging which allows in vivo assessment of their function as an implantable or percutaneous device. The authors tested 4- and 8-electrode devices in acute mouse (n=5) and pig (n=2) heart experiments. There are several limitations which reduce impact of this technology as presented.

1) Unfortunately, reproducibility of the presented animal data is impossible to evaluate. Figures show only single examples of the data without evaluation of the quality of electrograms, or averages from multiple experiments. The methods section mentions that the authors did 5 mouse and 2 pig acute experiments, but no comparison across different experiment are presented.

2) Low count of electrodes makes it difficult to establish feasibility of cardiac mapping with this technology. Basic and clinical electrophysiology mapping requires much higher count and better spatial resolution in order to map activation sequence of electrical activity during sinus rhythm, electrical pacing and during arrhythmia. No such analysis was done.

3) Presented single experiment with ischemia is a suspect. To produce infarction in a mouse model ligation is usually kept for 20-40 minutes (<https://doi.org/10.1152/ajpheart.00335.2017>). It is unusual to see ST segment changes after only 30 seconds of ischemia. Please present data from all 5 animals used in these experiments and show electrograms for different times up to 30 minutes. Please also present global ECG to illustrate onset of ischemia.

4) In general, quality of presented electrograms is low and not clear how reproducible it was from experiment to experiment. Figure S10 shows irregular rhythm in the mouse which suggests poor perfusion. Please present impedance measurements of electrodes for different frequencies. Importantly, electrograms recorded with these novel materials should be compared to state-of-the-art sensing electrodes using in a basic and/or clinical EP lab.

5) A possibility to create novel implantable devices with this poroelastic materials is particularly intriguing. Unfortunately, only histology was presented in Figure S13 which does not allow evaluation of inflammatory markers.

Specific comments:

Figure 1 illustrates the process of design and production of a poroelastic biosensor array. It is not clear how 3D information about cardiac anatomy obtained from the ultrasound imaging was actually used. It appears that the device is two-dimensional and conforms with the surface due to its flexibility. Please clarify.

Figure 2 presents comprehensive testing of mechanical material properties. However, electrical impedance measurements are missing. What is the frequency response of these materials and the devices, which combine poroelastic materials with gold electrodes? Electrogram recordings in the heart require high fidelity of sensing at wide range of frequencies.

Figure 3 presents what appears to be non-functional devices printed for different mammalian hearts: from piglet to bovine. Without electro-mechanical testing such data has limited usefulness.

Figure 4 illustrates recordings from 4 or 8 electrodes applied to mouse or pig hearts. Unfortunately, low temporal resolution of presented recordings (panel C) does not permit to evaluate their quality. Electrode arrays are usually used to map electrical activity in order to identify the source of arrhythmia. Unfortunately, the authors did not present activation maps at different conditions, i.e. paced versus sinus rhythm, versus arrhythmia, as did not quantify conduction velocity or other characteristics of electrical excitation of the heart. Another concern is

low resolution of the presented arrays: 4 and 8 electrodes only. In order to map electrical activity, one needs at least tens (30-60) electrodes. 64 or 256 electrodes are used in clinical mapping system. Was such low count determined by the low printing resolution of the technology? Or one can make poroelastic sensors with much higher spatial resolution? How many experiments did you conduct? Please indicate reproducibility of your results and quantify the quality of recordings.

Figure 5 illustrates recordings in the mouse heart during ligation of a coronary artery which causes ischemia and myocardial infarction. Again, resolution is very low, with 6 electrodes – which does not permit accurate localization of infarction versus healthy myocardium. Moreover, it is unusual to see dramatic difference in ST segment within 30 or 60 seconds of ischemia, because heart muscle has a reserve of metabolic substrates and ATP for few minutes. It takes minutes to observe such changes depending on the degree of obstruction of blood flow. Alternative explanation of this apparent change in ST segment of the ECG is possible motion artifacts due to movement of electrodes on the surface of the heart. Again, how many experiments did you do? And how reproducible were the results? Please quantify.

Reviewer #3 (Remarks to the Author):

The manuscript reports on the fabrication of soft poroelastic biosensor for simultaneous epicardial recording and imaging. It is an interesting paper, which can be publishable in nature communications. However, the following points should be addressed before publication.

- 1) The authors should compare their results with the existing methods for fabricating stretchable conductors. I recommend the use of table for comparative analysis. Performance of sensor, stretchability, and other figure of merits should be compared.
- 2) Biocompatibility. The authors should address the biocompatibility of their sensors. MTT assay test, Immune response, and other issues should be addressed.

Response to the Reviewer #1

We thank this reviewer for favorable comments, such as “Overall, this manuscript addresses an important challenge in the area of direct ink writing...”, “To the knowledge of this reviewer, the approach the authors is taking regarding the ink design, formation and fabrication is highly novel.”, and “The manuscript also has great depth and breadth regarding the studies of fundamental poroelastic materials science and demonstrations of important cardiac bio-electronic systems”, and the recommendation for publication in this journal.

Comment #1: On page 3, the unit for sheet resistance should be Ohm per square instead of just Ohm.

Our Responses: We thank the reviewer for this comment. We corrected the unit throughout the manuscript as well as in Supplementary Figure S1.

Comment #2: Both Ag and Cu are not used often in implantable devices due to their limited biocompatibility (as also demonstrated in their in vitro cell culturing studies), although in this manuscript they are covered by Au. Can the authors comment on the possibility of getting rid of them entirely from the system?

Our Responses: We thank the reviewer for this comment. In this work, we used Ag flakes as a seed for the subsequent electroless plating of Cu. The entire surface of Cu/Ag was then overcoated with Au to promote biocompatibility, which has been widely used for various biomedical applications. As a possible way of getting rid of Ag and Cu, the Cu-plated Ag flakes could be potentially replaced with Au-plated Au flakes for ensuring biocompatibility, despite their high costs (e.g., Au flakes: ~\$520 per gram). To clarify this aspect, we added the following texts, “Here, the Cu-plated Ag flakes were alternatively used as substitutes to expensive Au products, while their entire surface was plated with Au for 2 minutes to promote biocompatibility.²⁸” in page 3.

28. Vafaiee, M., Vossoughi, M., Mohammadpour, R. & Sasanpour, P. Gold-plated electrode with high scratch strength for electrophysiological recordings. *Sci. Rep.* **9**, 2985 (2019).

Comment #3: A related issue to the above is the adhesion of metal to the silicone porous substrate. From the R/R0 and strain test results and the cyclic stretching test the interface seems to be very stable. Can the authors perhaps elaborate on what leads to the stable interface between Ag and silicone?

Our Responses: We thank the reviewer for this question. Both the excellent adhesion of metal to the silicone porous substrate and their stable interface against cyclic stretching tests are attributed to the poroelastic nature of our device. For instance, our device contains densely networked conductive nanofillers that are embedded through the internal pores of the silicone porous substrate. This unique configuration of our device provides a little risk for delamination or separation of the embedded conductive nanofillers from the silicone porous substrate even under cyclic stretching tests. To highlight this aspect, we added the following texts, “... (3) reliable structural integrity in which conductive nanofillers are integrated through the internal pores of the sponge-like foam to minimize a risk of delamination or separation against cyclic deformations.” in page 2; and “Uniquely, the resulting devices are monolithic in which the densely networked conductive nanofillers are embedded through the internal pores in a way that eliminates a risk of compromising their structural integrity even under large deformations.” in page 8.

Comment #4: In addition to stretching test, the other important reliability measurement is soak test. The authors are highly recommended to look at the performance of their DIW electrodes as a function of soaking.

Our Responses: We thank the reviewer for this comment. As the reviewer suggested, we conducted a soak test with our DIW electrodes (1 cm × 3 cm) by immersing them in a bath of distilled (DI) water, phosphate buffered saline (PBS), and ethanol for 12 hours. The results show that the sheet resistance of our DIW electrodes remained nearly unchanged within a range of variation ($0.5 - 2.5 \Omega \cdot \text{sq}^{-1}$), confirming its long-term stability under wet environment. We summarized the key results in Supplementary Figure S7 and also added the following texts, “Moreover, the sheet resistance of the printed device remained nearly unchanged within a range of variation ($0.5-2.5 \Omega \cdot \text{sq}^{-1}$) when soaked in a bath of distilled (DI) water, phosphate buffered saline (PBS), and ethanol for 12 hours (Supplementary Figure S7).” in page 4.

Figure S7. Change in the sheet resistance of the devices soaked in a bath of DI water (top panel), PBS (middle panel), and ethanol (bottom panel) for 12 hours.

Comment #5: The 200 micron by 200 micron electrode from their direct ink writing is actually having remarkably small impedance of only 2.1 kilo Ohm at 40Hz. This is truly high performance, largely due to the porous structure in the electrodes.

Our Responses: We thank the reviewer again for this favorable comment. The poroelastic silicone composite provides not only large interfacial areas between electrodes and electrolyte but also rapid solid-state diffusion of charge carriers.³⁵ These aspects allowed the device to produce remarkably low electrochemical impedance as also shown in Supplementary Figure S10. To further highlight this aspect, we added the following texts, “The electrochemical impedance of the individual electrodes ($200 \mu\text{m} \times 200 \mu\text{m}$) was characterized in a phosphate-buffered saline (PBS) solution with a pH of 7.2 at 23 °C as 2.1, 1.5, and 1.0 k Ω at frequencies of 40, 150, and 1,000 Hz, respectively (Supplementary Figure S10). The remarkably low impedances were attributed to the poroelastic properties of the devices that provide not only large interfacial areas between electrodes and electrolyte but also rapid solid-state diffusion of charge carriers.³⁵” in page 5.

35. Jo, C., Park, Y., Jeong, J., Lee, K.T. & Lee, J. Structural effect on electrochemical performance of ordered porous carbon electrodes for na-ion batteries. *ACS Appl. Mater. Interfaces* 7, 11748-11754 (2015).

Figure S10. Electrochemical impedance of the device soaked in a bath of PBS as a function of frequency.

Response to the Reviewer #2

We thank this reviewer for favorable comments, such as "... potentially impactful new technology for stretchable biosensors that could be produced using direct ink writing approach.", "Presented poroelastic materials possess impressive mechanical properties, which allow such devices to conform to soft and mechanically active biological tissues, such as the heart.", "An additional benefit of these materials is their compatibility with ultrasound imaging which allows in vivo assessment of their function as an implantable or percutaneous device", and "A possibility to create novel implantable devices with this poroelastic materials is particularly intriguing".

Comment #1: Unfortunately, reproducibility of the presented animal data is impossible to evaluate. Figures show only single examples of the data without evaluation of the quality of electrograms, or averages from multiple experiments. The methods section mentions that the authors did 5 mouse and 2 pig acute experiments, but no comparison across different experiment are presented.

Our Responses: We thank the reviewer for this comment. In the revised manuscript, we showed all the measurement data obtained from 5 infarcted mice (i.e., total 9 recording channels), 5 healthy mice (i.e., total 9 recording channels), and 2 healthy pigs (i.e., total 12 recording channels), along with their quantitative evaluation. Supplementary Figure S13 displays the epicardial ECG signals obtained from 5 infarcted mice (i.e., 9 recording channels) before and 20, 30, 40, 50, and 60 seconds after ligation. The quantitative evaluations for the R-R interval, QRS duration, and J-point elevation of these signals are also included. In addition, Supplementary Figure S12 displays the epicardial ECG signals obtained from 5 healthy mouse (i.e., total 9 recording channels) and 2 healthy pigs (i.e., total 12 recording channels), along with the quantitative evaluation of R-R interval, QRS duration, and J-point elevation. Accordingly, we also added the following texts, "All the ECG data obtained from the infarcted mice (n = 5; total 9 recording channels) and their corresponding quantitative data of R-R interval, QRS duration, and J-point elevation are summarized in Supplementary Figure S13. The data exhibited the prolongation of QRS duration, the elevation of J-segment, the metrics of systolic dysfunction, and the elevation of ST-segment after about 40 seconds of ligation, showing statistical differences from those obtained before ligation using one-way ANOVA with Tukey's post hoc test with significance is set at $p < 0.05$." in page 6; "... in healthy murine (n = 5) and porcine hearts (n = 2) in vivo." in page 5; and "The raw data of the epicardial ECG signals obtained from all healthy murine and porcine hearts are summarized in Supplementary Figure S12a, displaying a typical ECG tracing of the cardiac cycle that consists of a P-wave (atrial depolarization), a QRS-complex (ventricular depolarization), and a T-wave (ventricular repolarization). The corresponding quantitative data of R-R interval, QRS duration, and J-point elevation are included in Supplementary Figure S12b, confirming that no J-segment elevation was observed in ECG recordings from the healthy hearts." in page 5.

Figure S13. (a) Epicardial ECG signals obtained from a murine myocardial infarction model (n = 5). (b) The corresponding quantitative data of R-R interval (left panel), QRS duration (middle panel), and J-point elevation (right panel).

Figure S12. (a) Epicardial ECG signals obtained from the healthy murine (n = 5) and porcine hearts (n = 2). (b) The corresponding quantitative data of R-R interval, QRS duration, and J-point elevation.

Comment #2: Low count of electrodes makes it difficult to establish feasibility of cardiac mapping with this technology. Basic and clinical electrophysiology mapping requires much higher count and better spatial resolution in order to map activation sequence of electrical activity during sinus rhythm, electrical pacing and during arrhythmia. No such analysis was done.

Our Responses: We thank the reviewer for this comment. For the in vivo studies, we incorporated total 8 electrodes (i.e., total 4 recording channels) and 16 electrodes (i.e., total 8 recording channels) in the devices to cover the left ventricular (LV) area of the murine and porcine hearts, respectively. The spatial resolution of the devices (i.e., number of electrodes within a given region) was determined by the minimum nozzle size (about 100 μm of inner diameter) for the nozzle injection system used in this study. To demonstrate higher count of electrodes, we included additional experimental results of a sensor array that contains total 64 electrodes (i.e., total 32 recording channels) on the enucleated porcine heart (similar size to the human heart) (Supplementary Figure S8). We also measured the epicardial ECG signals by applying an artificial ECG waveform (frequency = 1 Hz; amplitude = 20 mV) through the tissues of the enucleated porcine heart. Accordingly, we added the following texts, "... minimum inner diameter of nozzles = 100 μm ; ..." in page 2; "Here, the spatial resolution of these devices (i.e., the number of electrodes within a given region) was determined by the feature resolution of the nozzle injection system (i.e., the minimum nozzle size of about 100 μm). For instance, a total of 64 electrodes (i.e., 32 recording channels) on the device were uniformly distributed across the entire surface of the enucleated porcine heart that is similar in size to the human heart." in page 5; "Supplementary Figure

S8 presents representative epicardial ECG signals obtained from the enucleated porcine heart by applying an artificial ECG waveform (frequency = 1 Hz; amplitude = 20 mV) using a signal generator (Keithley 3390). The results showed that the measured ECG signals were followed consistently by those generated from the signal generator, confirming that all 64 electrodes seamlessly interfaced with the epicardial surface,” in page 5; and “The in vivo studies also suggest an opportunity to further increase the spatial resolution of these devices (i.e., the number of electrodes within a given region) in order to alleviate the need for reliance on post-processing algorithms to map the site of myocardial infarction in a higher resolution.” in page 8.

Figure S8. (a) Schematic illustration of an experiment setup for the ex vivo measurement of epicardial ECG signals from the enucleated porcine heart. (b) Photograph of the device (total 64 electrodes) placed on the surface of the enucleated porcine heart. (c) Measurement results of the epicardial ECG signals by applying an artificial ECG waveform.

Comment #3: Presented single experiment with ischemia is a suspect. To produce infarction in a mouse model ligation is usually kept for 20-40 minutes (<https://doi.org/10.1152/ajpheart.00335.2017>). It is unusual to see ST segment changes after only 30 seconds of ischemia. Please present data from all 5 animals used in these experiments and show electrograms for different times up to 30 minutes. Please also present global ECG to illustrate onset of ischemia.

Our Responses: We thank the reviewer for bringing up an important point regarding this murine infarction model. Confirmation of a successful ligation is critical in this murine infarction model where the left coronary artery can be difficult to visualize. Lindsey et al. also noted that “failure to induce MI can occur, usually due to missing the coronary artery during the ligation step. Monitoring the electrocardiogram for ST segment elevation during the procedure reduces this possibility”. [Am. J. Physiol. Heart Circ. Physiol. 314, H812-H838, 2018] In our experiments, we observed immediate changes in the ST-segment level, noted as either ST-elevation or -depression, within 40 seconds following coronary artery ligation that persisted or became more prominent over time. We also confirmed successful ligation by visible myocardial blanching in regions distal to the ligation site within minutes. This is a suggestive of acute ischemia in the myocardium rather than motion artifacts as it persists beyond the period of suture tightening. As the reviewer suggested, we included additional data showing the ECG signals for different times up to 30 minutes (Supplementary Figure S15). The results show that ST-segment elevation occurred in a murine model as early as 1-5 minutes post coronary artery ligation, which is consistent with many other previous reports.⁴⁰⁻⁴³ We also included additional data displaying reciprocal ST-depression and -elevation within seconds following left coronary artery ligation (Supplementary Figure S14) using a conventional 3-lead electrode recording configuration (control). As also suggested, we presented the epicardial ECG signals obtained from all 5 infarcted mice (i.e., total 9 recording channels) before and 20, 30, 40, 50, and 60 seconds after ligation (Supplementary Figure S13). The corresponding quantitative evaluations of these signals are also included. More details of our discussions about this aspect are shown in our responses to the Comment #1. In addition to these, we also added the following texts, “Figure 5c presents the results of control measurements obtained simultaneously using a 3-lead electrode set, displaying reciprocal ST-segment depression to confirm the occurrence of an ischemic event. The control measurements also displayed both reciprocal ST-segment depression and elevation within seconds of left coronary artery ligation (Supplementary Figure S14), which typically occurs in the 3-lead electrode recording configuration.⁴⁰⁻⁴³ These ST-segment depression and elevation were consistently observed during the surgery and throughout the recording duration of 30 minutes after ligation (Supplementary Figure S15).” in page 6.

Our Additional Comments: The reviewer correctly pointed out that the metabolic/ATP reservoir in the myocardium can lead to ST elevation typically on the order of minutes following coronary artery ligation. Although this observation is consistent in humans or large animal models, it is important to note that metabolism is significantly accelerated in mice like many other physiologic parameters (heart rate, infarct/cardiac remodeling time-course, etc.). [NPG Regen. Med. 4, 1-15, 2019] As such, the rapid depletion of these metabolic/ATP reservoir is often observed, which results in ST segment changes within seconds following ligation. Indeed, the publication by Lindsey et al. also noted that in mice, “coronary occlusion causes immediate cessation of aerobic metabolism in the ischemic myocardium, leading to rapid ATP depletion and metabolite accumulation and resulting in severe systolic dysfunction within seconds”. [Am. J. Physiol. Heart Circ. Physiol. 314, H812-H838, 2018] We of course acknowledge that an ischemic period of at least 30 minutes is necessary to induce irreversible changes that will lead to long-term damage and remodeling, as many studies have reported increasing infarct severity with longer ischemic period. [J. R. Soc. Interface, 16, 20190570, 2019]

40. Arora, N. & Mishra, B. Characterization of a low cost, automated and field deployable 2-lead myocardial infarction detection system. In: International Conference on COMMUNICATION SYSTEMS and NETWORKS (COMSNETS). (Bangalore, India, 2020).
41. Scofield, S.L.C. & Singh, K. Confirmation of myocardial ischemia and reperfusion inj

- ury in mice using surface pad electrocardiography. *J. Vis. Exp.* **117**, e54814 (2016).
42. Al-Salam, S. & Hashmi, S. Galectin-1 in early acute myocardial infarction. *PLOS ONE* **9**, e86994 (2014).
43. Preda, M.B. & Burlacu, A. Electrocardiography as a tool for validating myocardial ischemia-reperfusion procedures in mice. *Comp. Med.* **60**, 443-447 (2010).

Figure S14. Representative global ECG data of ST-segment depression (top panel) and elevation (bottom panel) within seconds following left coronary artery ligation (blue arrows).

Figure S15. Representative ECG data with persistent ST-segment elevation at baseline and at 1, 15, and 30 minutes following left coronary artery ligation.

Comment #4: In general, quality of presented electrograms is low and not clear how reproducible it

was from experiment to experiment. Figure S10 shows irregular rhythm in the mouse which suggests poor perfusion. Please preset impedance measurements of electrodes for different frequencies. Importantly, electrograms recorded with these novel materials should be compared to state-of-the-art sensing electrodes using in a basic and/or clinical EP lab.

Our Responses: We thank the reviewer for this comment. In the revised Supplementary Figure S12, we replaced the ECG data with better quality ones, displaying regular rhythms. In addition, the revised Supplementary Figure S10 shows the frequency response of our sensor soaked in a phosphate-buffered saline (PBS) with pH = 7.2 at 23 °C. Notably, our sensor provided remarkably low electrochemical impedances of 2.1 k Ω , 1.5 k Ω and 1.0 k Ω at the frequency of 40 Hz, 150 Hz, and 1 kHz, respectively. We also added more ECG data of control measurements obtained from a 3-lead electrode setup (SA instrument) that is a state-of-the-art in vivo sensing tool tailored for small animal applications (Supplementary Figures S14 & S15). More details of our discussions about these are shown in our responses to the Comments #1 & #3. Accordingly, we added the following texts, “The electrochemical impedance of the individual electrodes (200 $\mu\text{m} \times 200 \mu\text{m}$) was characterized in a phosphate-buffered saline (PBS) solution with a pH of 7.2 at 23 °C as 2.1, 1.5, and 1.0 k Ω at frequencies of 40, 150, and 1,000 Hz, respectively (Supplementary Figure S10). The remarkably low impedances were attributed to the poroelastic properties of the devices that provide not only large interfacial areas between electrodes and electrolyte but also rapid solid-state diffusion of charge carriers.³⁵” in page 5; and “For comparison, Figure 4c (bottom panel) presents the results of the control measurements simultaneously obtained from a 3-lead electrode set (ERT Control/Gating Module Model 1030, SA Instruments, Stony Brook, NY) that presents a state-of-the-art in vivo sensing tool tailored for small animal applications.” in page 6.

Our Additional Comments: Electroanatomical mapping - commonly used in the clinic to identify sources of arrhythmias for ablation therapy - has not been successfully performed in mice due to the small size and fast heart rate of the murine heart. This approach is also challenging in rodents as it requires the use of an intracardiac catheter. Further, these methodologies require highly specialized equipment/tools. The majority of in vivo ECG recording in mice has been through the use of surface or needle electrodes, often in a telemetry setup. In this study, we used a 3-lead electrode setup (SA instrument) as it is one of the state-of-the-art in vivo sensing tools tailored for small animal applications. As schematically illustrated in Figure 5d, our measurement setup allowed us to record epicardial ECG signals simultaneously using both our sensor and the 3-lead electrodes in mice during/after coronary artery ligation surgeries.

35. Jo, C., Park, Y., Jeong, J., Lee, K.T. & Lee, J. Structural effect on electrochemical performance of ordered porous carbon electrodes for na-ion batteries. *ACS Appl. Mater. Interfaces* 7, 11748-11754 (2015).

Figure S10. Electrochemical impedance of the device soaked in a bath of PBS as a function of frequency. **Comment #5:** A possibility to create novel implantable devices with this poroelastic materials is particularly intriguing. Unfortunately, only histology was presented in Figure S13 which does not allow evaluation of inflammatory markers.

Our Responses: We agree with this reviewer that our manuscript would be benefited from a more detailed and thorough discussion of this topic. In this study, we used both H&E and Masson’s Trichrome stains as they provide basic insights into inflammatory responses, such as the invasion of eosinophils, neutrophils, macrophages, and multinucleated giant cells, while allowing the identification of peripheral lymphocytes, plasma cells, fibroblasts, and fibrous connective tissue leading towards fibrotic encapsulation. Detailed discussions of the histological analysis are summarized in the Methods section. In addition, we also added a number of photomicrographs in Supplementary Figures S18-20 to provide more detailed information about stained myocardium with annotations pointing out specific cell-types and increased epicardial thickening near and far away from the implant site under the guidance from a board-certified veterinary pathologist (Prof. Abigail Cox in Purdue College of Veterinary Medicine – co-author of this manuscript). Specifically, Supplementary Figure S18a provides an overview of granuloma and aorta on day 7 post-implant. Supplementary Figures S18b and c demonstrate the composition of granuloma, as well as macrophages, multi-nucleated giant cells, and neutrophils directed at the implanted device. Supplementary Figure S19 presents a comparison of thickening of epicardium near the implanted device, showing a progression from $44.4 \pm 8.3 \mu\text{m}$ to $645.9 \pm 5.3 \mu\text{m}$ in thickness, which demonstrates the development of foreign body response. Supplementary Figure S20 includes the chronic epicarditis on days 7 and 14 post-implants for pathological evaluation. Accordingly, we added the following texts, “Evaluation of the in vivo biocompatibility and anti-biofouling properties of the custom-printed devices and their effect on cardiac function is a critical factor for demonstrating their long-term engraftment.” in page 7; “Supplementary Figure S18 provides an overview of nearby granuloma and aorta on day 7 post-implant of the device. The magnified views of granuloma, macrophages, and multi-nucleated giant cells at the surface of the implanted device suggest chronic inflammation. Supplementary Figure S19 presents an increased thickening of epicardium near the implanted device on days 1, 7, and 14 post-implants, showing its progression from $44.4 \pm 8.3 \mu\text{m}$ to $645.9 \pm 5.3 \mu\text{m}$ in thickness. The results indicate that chronic inflammatory response directed towards the implanted device. The corresponding chronic epicarditis on days 7 and 14 post-implants for pathological evaluation are shown in Supplementary Figure S20. Detailed discussions of the histological analysis are also summarized in the Methods section.” in page 7; and “The stained tissues were imaged in segments at 10 \times , 20 \times , and 40 \times magnification using a Leica ICC50W stereomicroscope (Leica Microsystems Inc., Buffalo Grove, IL), and stitched together with MosaicJ. Epicardial thickening was measured from histological images using ImageJ. Microscopic examination was performed by a board-certified veterinary pathologist and the interpretation was based on standard histopathological morphology of a murine heart. Four serial transverse sections of the heart from the base to the mid-papillary region were histologically evaluated. The results revealed that, on day 7 post-implant, the implanted device was present adjacent to the wall of aorta invoking the formation of a granuloma (Supplementary Figure 18). The granuloma was comprised of a central necrotic core of eosinophilic cellular and karyorrhectic debris mixed with numerous degenerate and viable neutrophils. Surrounding the necrotic core were epithelioid macrophages and multinucleated giant cells, rimmed peripherally by lymphocytes, plasma cells, fibroblasts, and fibrous connective tissue. The epicardium of the right and left atrium was expanded by fibroblasts admixed by neutrophils, lymphocytes and macrophages (Supplementary Figure 19). The fragments of the foreign materials surrounded by inflammatory cells were found adjacent to the wall of the right atrium. On day 14 post-implant, the epicardium of the right ventricle was thickened with pale eosinophilic collagen fibers and increased fibroblast cellularity. The pericardium was similarly thickened with collagen, numerous fibroblasts, small caliber blood vessels and infiltrates of lymphocytes, plasma cells, macrophages, multinucleate giant cells and few neutrophils. The thickened pericardium surrounded fragments of granular, black foreign material with multifocal adhesions to the underlying epicardium (Supplementary Figure 20). The observed lesions predominantly bordered the right ventricular free wall. The pericardium was only observed on the right-side following tissue processing, likely because the pericardium was adhered to the underlying inflamed epicardium. Although chronic inflammation due to a foreign body response was present at the implant site after long-term implantation, its effects on intraoperative epicardial mapping or cardiac ejection

fraction were insignificant. While short-term intraoperative implantation side effects cannot be ruled out, we observed no issues during intraoperative epicardial mapping or cardiac ejection fraction. The inflammation could be further reduced through the inclusion of anti-biofouling surfacing coatings or textures.^{46,51,52} in page 14.

Our Additional Comments: Based on our MTT assay using heart myoblasts (H9C2) cells and BSA-FITC protein adhesion evaluation, we expect that the Au-coated sensors are non-cytotoxic and do not encourage protein aggregation at the implant site compared to other similar implantable sensors. Further, while chronic inflammation due to a foreign body response was present on day 14 post-implant, the effect of the implanted device on cardiac function was negligible based on the minimal changes to an ejection fraction of the heart during the entire implantation period on the murine epicardial surface (Figure 6e and 6f, Supplementary Movie S5). The results also show that the ejection fraction of the heart with the implant (60-70%) was clearly higher than the abnormal ranges for ischemia-reperfusion injury (40-60%) or permanent ligation (20-40%). Thus, we believe that the inflammation caused by our device has a negligible effect on global cardiac performance.

46. Xu, J.J., Xu, J., Moon, H., Sintim, H.O. & Lee, H. Zwitterionic porous conjugated polymers as a versatile platform for antibiofouling implantable bioelectronics. *ACS Appl. Polym. Mater.* **2**, 528-536 (2020).
51. Xu, J., Xu, J., Moon, H., Sintim, H.O. & Lee, H. Zwitterionic liquid crystalline polythiophene as an antibiofouling biomaterial. *J. Mater. Chem. B* **9**, 349-356 (2021).
52. Xu, J. et al. One-step large-scale nanotexturing of nonplanar ptfe surfaces to induce bactericidal and anti-inflammatory properties. *ACS Appl. Mater. Interfaces* **12**, 26893-26904 (2020).

Figure S18. (a) Overview of granuloma and aorta on day 7 post-implant. (b) Magnified views (at 20 \times) of granuloma formation on day 7 post-implant. The boxed area identifies the presence of macrophages, multi-nucleated giant cells, and fibroblasts. The inset image demonstrates macrophages containing phagocytized debris resulting from the intralésional device. (c) Magnified views (20 \times) of macrophages and multi-nucleated giant cells at the surface of the implanted device (red arrows). The neutrophils (blue arrows) suggest acute to chronic inflammation directed at the device.

Figure S19. (a) Day 1 epicardium at 20 × (left) and 40 × (right) magnification. The epicardium presents at five-times normal thickness and is comprised of neutrophils (red arrows). (b) Day 7 epicardium. The double-headed line indicates a thickening of epicardium by mononuclear cells and fibroblasts due to chronic inflammatory response. (c) Day 14 epicardium. The double-headed line indicates a worsening of epicardial thickening. The presence and epithelioid morphology of macrophages (red arrow) indicate chronic granulomatous response (e.g., foreign body response) at the surface of the implanted device. The neutrophils (blue arrow) suggests that acute chronic inflammation directed towards the implanted device. (d) Progression of epicardial thickness measured using ImageJ (n = 5). The measurements were spaced at 100 μm perpendicularly from the epicardial surface to the underlying muscle layer.

Figure S20. (a) Chronic epicarditis observed on day 7 post-implant. (b) Pericardial to epicardial adhesion with intralésional device observed on day 14 post-implant.

[Specific Comments]

Specific Comment #1: Figure 1 illustrates the process of design and production of a poroelastic biosensor array. It is not clear how 3D information about cardiac anatomy obtained from the ultrasound imaging was actually used. It appears that the device is two-dimensional and conforms with the surface due to its flexibility. Please clarify.

Our Responses: We thank the reviewer for this comment. The 3D geometric information of the heart obtained from ultrasound images provide important insights into the overall size, geometry, and structure of the infarcted area. Therefore, the information was taken into consideration to precisely scale and adjust scale and adjust the layout of our sensor in order to meet the requirement of a specific geometric accuracy such that the distance between the pairs of recording electrodes are well aligned to the position and orientation of the infarcted area. To clarify this, we added the following texts, “This 3D geometry was taken into consideration to precisely scale, adjust, and tailor the overall layout of the device to meet the requirement of a specific geometric accuracy. This aspect allowed the device to seamlessly cover the entire infarcted area when placed on the curvilinear epicardial surface of the heart in a manner that enabled the accurate alignment of the embedded recording electrodes to the position and orientation of the infarcted area.” in page 2.

Specific Comment #2: Figure 2 presents comprehensive testing of mechanical material properties. However, electrical impedance measurements are missing. What is the frequency response of these materials and the devices, which combine poroelastic materials with gold electrodes? Electrogram recordings in the heart require high fidelity of sensing at wide range of frequencies.

Our Responses: We thank the reviewer for this comment. Supplementary Figure S10 shows the frequency response of our sensor soaked in a phosphate-buffered saline (PBS) with pH = 7.2 at 23 °C. Notably, our sensor provided remarkably low electrochemical impedances of 2.1 k Ω , 1.5 k Ω and 1.0 k Ω at the frequency of 40 Hz, 150 Hz, and 1 kHz, respectively. This is attributed to the poroelastic nature of our sensor that provides not only large interfacial areas between electrodes and electrolyte but also rapid solid-state diffusion of charge carriers. More details of our discussions about this aspect are shown in our responses to the Comment #4.

Specific comment #3: Figure 3 presents what appears to be non-functional devices printed for different mammalian hearts: from piglet to bovine. Without electro-mechanical testing such data has limited usefulness.

Our Responses: We thank the reviewer for this comment. Figure 3 highlights a key aspect of our optimized ink formulations and high precision direct-ink-writing (DIW) method, which is capable of generating tailored sensors to accommodate the different sizes and shapes of the enucleated piglet, ovine, porcine, and bovine hearts. As the reviewer suggested, we demonstrated the electro-mechanical testing ex vivo by using a device that contains total 64 electrodes (i.e., total 32 recording channels) on the enucleated porcine heart (Supplementary Figure S8). We applied an artificial ECG waveform (frequency = 1 Hz; amplitude = 20 mV) through the tissue of the enucleated porcine heart using a signal generator (Keithley 3390). The results showed that the recorded ECG signals were nearly identical to those generated from the signal generator. More details of our discussions about this aspect are shown in our response to the Comment #2.

Specific Comment #4: Figure 4 illustrates recordings from 4 or 8 electrodes applied to mouse or pig hearts. Unfortunately, low temporal resolution of presented recordings (panel C) does not permit to evaluate their quality. Electrode arrays are usually used to map electrical activity in order to identify the source of arrhythmia. Unfortunately, the authors did not present activation maps at different conditions, i.e. paced versus sinus rhythm, versus arrhythmia, as did not quantify conduction velocity or other characteristics of electrical excitation of the heart. Another concern is low resolution of the presented

arrays: 4 and 8 electrodes only. In order to map electrical activity, one needs at least tens (30-60) electrodes. 64 or 256 electrodes are used in clinical mapping system. Was such low count determined by the low printing resolution of the technology? Or one can make poroelastic sensors with much higher spatial resolution? How many experiments did you conduct? Please indicate reproducibility of your results and quantify the quality of recordings.

Our Responses: We thank the reviewer for these questions. As suggested, we modified Figure 4c for better temporal resolution. In addition, we were also able to generate voltage map and identify the abnormality by acute ST-segment elevation (Figure 5f & Supplementary Movie S4), which was built upon our acute infarction studies in a murine myocardial infarction model (n = 5; total 9 recording channels) (Supplementary Figure S13). More details of our discussions about this aspect are shown in our response to the Comment #1. The spatial resolution (i.e., number of electrodes within a given size) was determined by the minimum nozzle size (about 100 μm of inner diameter) for the nozzle injection system used in this study. Higher spatial resolution can be potentially achieved by using a better resolution nozzle injection tool. To demonstrate higher count of electrodes, we included additional experimental results of a sensor array that contains total 64 electrodes (i.e., total 32 recording channels) on the enucleated porcine heart (similar size to the human heart) (Supplementary Figure S8). We also measured the epicardial ECG signals by applying an artificial ECG waveform (frequency = 1 Hz; amplitude = 20 mV) through the tissues of the enucleated porcine heart. In the revised manuscript, we showed all the measurement data obtained from 5 infarcted mice (i.e., total 9 recording channels), 5 healthy mice (i.e., total 9 recording channels), and 2 healthy pigs (i.e., total 12 recording channels), along with their quantitative evaluations. More details of our discussions about these aspects are shown in our response to the Comment #2.

Figure 4. (c) Simultaneously measured ECG signals using the custom-printed sensor array (top panel) and a control 3-lead electrode set (bottom panel) on a murine heart.

Figure 5. (f) Post-processed 3D image reconstructed from the spatiotemporally recorded epicardial ECG and ultrasound signals after 60 seconds post-ligation.

Specific Comment #5: Figure 5 illustrates recordings in the mouse heart during ligation of a coronary artery which causes ischemia and myocardial infarction. Again, resolution is very low, with 6 electrodes – which does not permit accurate localization of infarction versus healthy myocardium. Moreover, it is unusual to see dramatic difference in ST segment within 30 or 60 seconds of ischemia, because heart muscle has a reserve of metabolic substrates and ATP for few minutes. It takes minutes to observe such changes depending on the degree of obstruction of blood flow. Alternative explanation of this apparent change in ST segment of the ECG is possible motion artifacts due to movement of electrodes on the surface of the heart. Again, how many experiments did you do? And how reproducible were the results? Please quantify.

Our Responses: We thank the reviewer for these questions. As described in our response to the Comment #2 and Specific Comment #4, the spatial resolution (i.e., number of electrodes within a given size) was determined by the minimum nozzle size (about 100 μm of inner diameter) for the nozzle injection system used in this study. Higher spatial resolution can be potentially achieved by using a better resolution nozzle injection tool. Also, the rapid ST segment elevation in a murine model observed in our study is consistent to the results of many previous reports.⁴⁰⁻⁴³ The simultaneous ECG recordings obtained from our sensor and the control 3-lead electrodes confirmed successful ligation by visible myocardial blanching in regions distal to the ligation site within minute. More details of our discussions about this aspect are shown in our responses to the Comment #3. In the revised manuscript, we showed all the measurement data obtained from 5 infarcted mice (i.e., total 9 recording channels), 5 healthy mice (i.e., total 9 recording channels), and 2 healthy pigs (i.e., total 12 recording channels), along with their quantitative evaluations. In addition to these, we also added the following texts, “Successful ligation was confirmed by discoloration of the myocardium in regions distal to the ligation site and global ST segment elevation or depression in the 3-lead electrode set.” in page 12.

Response to the Reviewer #3

We thank this reviewer for favorable comments, such as “The manuscript reports on the fabrication of soft poroelastic biosensor for simultaneous epicardial recording and imaging. It is an interesting paper, which can be publishable in nature communications”. and the recommendation for publication in this journal.

Comment #1: The authors should compare their results with the existing methods for fabricating stretchable conductors. I recommend the use of table for comparative analysis. Performance of sensor, stretchability, and other figure of merits should be compared.

Our Responses: We thank the reviewer for this suggestion. We added a table to compare the key mechanical and electrical properties of our device materials with other existing materials (Table S1). Accordingly, we also added the following texts, “The comparison of this poroelastic silicone composite with other existing materials in terms of the mechanical and electrical properties is shown in Table S1.” in page 2.

Patterning Process	Materials	Conductivity	Maximum Stretchability	Young's Modulus	Ref
Direct Ink Writing	Poroelastic Biosensor (This Work)	$7.72 \pm 1.52 \Omega \cdot \text{sq}^{-1}$	150%	0.15 MPa	
	Ag Flakes/PEO	$13,800 \text{ S} \cdot \text{cm}^{-1}$	300 %	0.4 MPa	2
	CB/TPU	$0.841 \text{ S} \cdot \text{cm}^{-1}$	-	8.8 MPa	3
	Ag/PA	$15,200 \text{ S} \cdot \text{cm}^{-1}$	-	-	
	Ag Flakes/TPU	$10^4 \text{ S} \cdot \text{cm}^{-1}$	240%	2.3 MPa	7
	Ag/Dragon Skin	500 Ω	250%	0.8 MPa	17
Moulding	Ag NWs/SBS	$11,210 \text{ S} \cdot \text{cm}^{-1}$	50%	40 MPa	24
	Ag-Au NWs/SBS	$72,600 \text{ S} \cdot \text{cm}^{-1}$	840%	37.4 MPa	32
Photolithography	Graphene-Ag NWs/PDMS	$33 \Omega \cdot \text{sq}^{-1}$	100%	-	9
	Ag NWs/PDMS	$26.1 \Omega \cdot \text{sq}^{-1}$	73%	-	10
	Au/Ni/PDMS	$1.9 \Omega \cdot \text{sq}^{-1}$	80%	-	25
Screen Printing	Ag Flakes/Fluoroelastomer	$0.06 \Omega \cdot \text{sq}^{-1}$	450%	-	5
	CNTs/Fluorinated rubber	$57 \text{ S} \cdot \text{cm}^{-1}$	134%	-	21
Mask Patterning	Ag NWs/PDMS	7.5 Ω	70%	6.32 MPa	11
	Au-TiO ₂ NWs/PDMS	$0.63 \Omega \cdot \text{sq}^{-1}$	100%	-	12
	Ag Flakes/PDMS	$5,695 \text{ S} \cdot \text{cm}^{-1}$	80%	-	13

Table S1. Comparisons of the poroelastic silicone composite with other existing materials in terms of the mechanical and electrical properties. The following abbreviations are used in this table. Nanowire (NW); Carbon Nanotube (CNT); Poly(Ethylene Oxide) (PEO); Carbon Black (CB); Thermoplastic PolyUrethane (TPU); Styrene-Butadiene-Styrene (SBS); Polydimethylsiloxane (PDMS)

- Zhu, Z. et al. 3D printed functional and biological materials on moving freeform surfaces. *Adv. Mater.* **30**, 1707495 (2018).
- Lind, J.U. et al. Instrumented cardiac microphysiological devices via multimaterial three-dimensional printing. *Nat. Mater.* **16**, 303-308 (2017).

5. Jin, H. et al. Enhancing the performance of stretchable conductors for e-textiles by controlled ink permeation. *Adv. Mater.* **29**, 1605848 (2017).
7. Valentine, A.D. et al. Hybrid 3D printing of soft electronics. *Adv. Mater.* **29**, 1703817 (2017).
9. Lee, M.-S. et al. High-performance, transparent, and stretchable electrodes using graphene–metal nanowire hybrid structures. *Nano Lett.* **13**, 2814-2821 (2013).
10. Park, S.-M. et al. Metal nanowire percolation micro-grids embedded in elastomers for stretchable and transparent conductors. *J. Mater. Chem. C* **3**, 8241-8247 (2015).
11. Amjadi, M., Pichitpajongkit, A., Lee, S., Ryu, S. & Park, I. Highly stretchable and sensitive strain sensor based on ag nws-elastomer nanocomposite. *ACS nano* **8**, 5154–5163 (2014).
12. Tybrandt, K. et al. High-density stretchable electrode grids for chronic neural recording. *Adv. Mater.* **30**, 1706520 (2018).
13. Kim, I. et al. A photonic sintering derived ag flake/nanoparticle-based highly sensitive stretchable strain sensor for human motion monitoring. *Nanoscale* **10**, 7890-7897 (2018).
17. Guo, S.Z., Qiu, K.Y., Meng, F.B., Park, S.H. & McAlpine, M.C. 3D printed stretchable tactile sensors. *Adv. Mater.* **29**, 1-8 (2017).
21. Sekitani, T. et al. A rubberlike stretchable active matrix using elastic conductors. *Science* **321**, 1468-1472 (2008).
24. Park, J. et al. Electromechanical cardioplasty using a wrapped elasto-conductive epicardial mesh. *Sci. Transl. Med.* **8**, 344ra386 (2016).
25. Jeong, G.S. et al. Solderable and electroplatable flexible electronic circuit on a porous stretchable elastomer. *Nat. Commun.* **3**, 977 (2012).
32. Choi, S. et al. Highly conductive, stretchable and biocompatible ag-au core-sheath nanowire composite for wearable and implantable bioelectronics. *Nat. Nanotechnol.* **13**, 1048-1056 (2018).

Comment #2: The authors should address the biocompatibility of their sensors. MTT assay test, Immune response, and other issues should be addressed.

Our Responses: We thank the reviewer for this comment. We agree with the reviewer that robust evaluation of the biocompatibility of our device is important for its translation potential as an implantable device. Therefore, we included the results of in vitro cell toxicity tests using an MTT assay on heart myoblast (H9C2) cells (Figure 6a), showing consistent cell proliferation throughout the assay period (24 hours) with no significant differences compared to control and bare sponge-like foam. As a negative control, our device without the Au overcoat showed reduced cell viability and cell toxicity. We also analyzed the resistance of our device to protein adhesion with BSA-FITC to characterize the anti-biofouling properties of the sponge-like microporous structure (Figures 6c-d). Taken together, these in vitro biocompatibility tests demonstrate that our device is non-cytotoxic and has improved protein adhesion properties compared to similar devices. In addition to these, we also added a number of photomicrographs in Supplementary Figures S18-20 to provide more detailed information about stained myocardium with annotations pointing out specific cell-types and increased epicardial thickening near and far away from the implant site under the guidance from a board-certified veterinary pathologist (Prof. Abigail Cox in Purdue College of Veterinary Medicine – co-author of this manuscript). Specifically, Supplementary Figure S18a provides an overview of granuloma and aorta on day 7 post-implant. Supplementary Figures S18b and S18c demonstrate the composition of granuloma, as well as macrophages, multi-nucleated giant cells, and neutrophils directed at the implanted device. Supplementary Figure S19 presents a comparison of thickening of epicardium near the implanted device, showing a progression from $44.4 \pm 8.3 \mu\text{m}$ to $645.9 \pm 5.3 \mu\text{m}$ in thickness, which demonstrates the development of foreign body response. Supplementary Figure S20 includes the chronic epicarditis on days 7 and 14 post-implants for pathological evaluation. Accordingly, we added the following texts, “Evaluation of the in vivo biocompatibility and anti-biofouling properties of the custom-printed devices and their effect on cardiac function is a critical factor for demonstrating their long-term engraftment.” in page 7; “Supplementary Figure S18 provides an overview of nearby granuloma and aorta on day 7

post-implant of the device. The magnified views of granuloma, macrophages, and multi-nucleated giant cells at the surface of the implanted device suggest chronic inflammation. Supplementary Figure S19 presents an increased thickening of epicardium near the implanted device on days 1, 7, and 14 post-implants, showing its progression from $44.4 \pm 8.3 \mu\text{m}$ to $645.9 \pm 5.3 \mu\text{m}$ in thickness. The results indicate that chronic inflammatory response directed towards the implanted device. The corresponding chronic epicarditis on days 7 and 14 post-implants for pathological evaluation are shown in Supplementary Figure S20. Detailed discussions of the histological analysis are also summarized in the Methods section.” in page 7; and “The stained tissues were imaged in segments at 10 \times , 20 \times , and 40 \times magnification using a Leica ICC50W stereomicroscope (Leica Microsystems Inc., Buffalo Grove, IL), and stitched together with MosaicJ. Epicardial thickening was measured from histological images using ImageJ. Microscopic examination was performed by a board-certified veterinary pathologist and the interpretation was based on standard histopathological morphology of a murine heart. Four serial transverse sections of the heart from the base to the mid-papillary region were histologically evaluated. The results revealed that, on day 7 post-implant, the implanted device was present adjacent to the wall of aorta invoking the formation of a granuloma (Supplementary Figure 18). The granuloma was comprised of a central necrotic core of eosinophilic cellular and karyorrhectic debris mixed with numerous degenerate and viable neutrophils. Surrounding the necrotic core were epithelioid macrophages and multinucleated giant cells, rimmed peripherally by lymphocytes, plasma cells, fibroblasts, and fibrous connective tissue. The epicardium of the right and left atrium was expanded by fibroblasts admixed by neutrophils, lymphocytes and macrophages (Supplementary Figure19). The fragments of the foreign materials surrounded by inflammatory cells were found adjacent to the wall of the right atrium. On day 14 post-implant, the epicardium of the right ventricle was thickened with pale eosinophilic collagen fibers and increased fibroblast cellularity. The pericardium was similarly thickened with collagen, numerous fibroblasts, small caliber blood vessels and infiltrates of lymphocytes, plasma cells, macrophages, multinucleate giant cells and few neutrophils. The thickened pericardium surrounded fragments of granular, black foreign material with multifocal adhesions to the underlying epicardium (Supplementary Figure 20). The observed lesions predominantly bordered the right ventricular free wall. The pericardium was only observed on the right-side following tissue processing, likely because the pericardium was adhered to the underlying inflamed epicardium. Although chronic inflammation due to a foreign body response was present at the implant site after long-term implantation, its effects on intraoperative epicardial mapping or cardiac ejection fraction were insignificant. While short-term intraoperative implantation side effects cannot be ruled out, we observed no issues during intraoperative epicardial mapping or cardiac ejection fraction. The inflammation could be further reduced through the inclusion of anti-biofouling surfacing coatings or textures.^{46,51,52”} in page 13.

Our Additional Comments: Based on our MTT assay using heart myoblasts (H9C2) cells and BSA-FITC protein adhesion evaluation, we expect that the Au-coated sensors are non-cytotoxic and do not encourage protein aggregation at the implant site compared to other similar implantable sensors. Further, while chronic inflammation due to a foreign body response was present on day 14 post-implant, the effect of the implanted device on cardiac function was negligible based on the minimal changes to an ejection fraction of the heart during the entire implantation period on the murine epicardial surface (Figure 6e and 6f, Supplementary Movie S5). The results also show that the ejection fraction of the heart with the implant (60-70%) was clearly higher than the abnormal ranges for ischemia-reperfusion injury (40-60%) or permanent ligation (20-40%). Thus, we believe that the inflammation caused by our device has a negligible effect on global cardiac performance.

46. Xu, J., Xu, J., Moon, H., Sintim, H.O. & Lee, H. Zwitterionic porous conjugated polymers as a versatile platform for antibiofouling implantable bioelectronics. *ACS Appl. Polym. Mater.* **2**, 528-536 (2020).
51. Xu, J., Xu, J., Moon, H., Sintim, H.O. & Lee, H. Zwitterionic liquid crystalline polythiophene as an antibiofouling biomaterial. *J. Mater. Chem. B* **9**, 349-356 (2021).

52. Xu, J. et al. One-step large-scale nanotexturing of nonplanar ptfе surfaces to induce bactericidal and anti-inflammatory properties. *ACS Appl. Mater. Interfaces* **12**, 26893-26904 (2020).

Figure S18. (a) Overview of granuloma and aorta on day 7 post-implant. (b) Magnified views (at 20 \times) of granuloma formation on day 7 post-implant. The boxed area identifies the presence of macrophages, multi-nucleated giant cells, and fibroblasts. The inset image demonstrates macrophages containing phagocytized debris resulting from the intralésional device. (c) Magnified views (20 \times) of macrophages and multi-nucleated giant cells at the surface of the implanted device (red arrows). The neutrophils (blue arrows) suggest acute to chronic inflammation directed at the device.

Figure S19. (a) Day 1 epicardium at 20 × (left) and 40 × (right) magnification. The epicardium presents at five-times normal thickness and is comprised of neutrophils (red arrows). (b) Day 7 epicardium. The double-headed line indicates a thickening of epicardium by mononuclear cells and fibroblasts due to chronic inflammatory response. (c) Day 14 epicardium. The double-headed line indicates a worsening of epicardial thickening. The presence and epithelioid morphology of macrophages (red arrow) indicate chronic granulomatous response (e.g., foreign body response) at the surface of the implanted device. The neutrophils (blue arrow) suggests that acute chronic inflammation directed towards the implanted device. (d) Progression of epicardial thickness measured using ImageJ (n = 5). The measurements were spaced at 100 μm perpendicularly from the epicardial surface to the underlying muscle layer.

Figure S20. (a) Chronic epicarditis observed on day 7 post-implant. (b) Pericardial to epicardial adhesion with intralesional device observed on day 14 post-implant.

REVIEWER COMMENTS

Reviewer #1 (Remarks to the Author):

The authors have fully addressed this reviewer's prior comments. This manuscript is ready for publish as is on Nature Communications.

Reviewer #3 (Remarks to the Author):

I carefully examined the changes in the revised manuscript and found that the manuscript can be published without further change. The authors addressed all the points raised by the reviewers and the manuscript contains the enough novelty publishable in Nature Communications.

Reviewer #4 (Remarks to the Author):

The authors have addressed point-by-point responses to the reviewer's comments. In addition, the manuscript is well revised accordingly. Overall, the reviewer believes the manuscript has substantially improved. However, the reviewer still has the following concerns for this work as well as some points of the author's responses.

Comments to the author's responses.

1.Regarding the response to reviewer's comment #1, it is important to compare the values such as R-R interval, QRS duration, and j-point presented in Figure S12 in the revised manuscript to those by typical values from the traditional test (control), which help to give an idea whether ECG data is distorted or not during the in-vivo experiment by using the soft poroelastic biosensor.

2.Although the author has added Figure S8 for spatial mapping using 64 electrodes as responses to reviewer's comment #2 and specific comment #5, the reviewer believes that the work definitely needs spatiotemporal electrophysiological mapping results to prove the significant impact of the biosensor on cardiac physiology.

3.The author emphasized the limitation of previous ink in terms of long-term reliable recording. Although the presented device itself is stable, it causes a chronic inflammatory response, as the author responds to comment #5. Therefore, this biosensor still has a long-term practical issue for implantable electronics.

4.The author mentioned anti-bio-fouling surface coating or texture as a potential solution for inflammatory response. However, the additional coating on the device causes the interlayer between the sensor and epicardium, which may cause high impedance. In addition, a textured device may also cause non-conformal contact between the sensor and epicardium. The reviewer believes the author should provide a more feasible solution.

5.Regarding specific comment #3, the author applied voltage function to the porcine heart (Figure S8). The reviewer believes those experiments may lead to the heart's electromechanical movement even though it was an ex-vivo experiment. Did the author observe such movement? If so, it will be helpful to quantify it to improve the quality of this work.

6.The author responded to specific comment #4 by describing Figure 5f as a spatial mapping recorded epicardial ECG from the murine heart. However, the result shows some voltage values (yellow and green) outside of the area covered by the sensor array. Could the author clarify these?

7.The author specified, "we showed all the measurement data obtained from 5 infarcted mice (i.e., total 9 recording channels) ..." in response to the specific comment #5. Figure 5a shows 6 sensing channels on the murine heart. If the author did the same experiment with 5 different infarcted mice, the total recording channel might not be 9. The author needs to clarify.

Additional comments on the work.

1.The author used PVA as a temporary supporting layer to deliver the sensor to the hearts. Although the author used warm saline solution to dissolve PVA, the PVA is not chemically decomposed by water, but it could be physically dissolved. Therefore, the PVA may remain just as it is in the interior of the body, which could cause some issues. Although PVA is generally considered biocompatible regarding oral intake, contact to the interior of the body and oral intake are considered different in terms of toxicity. The reviewer has concerns about it for practical usage of this work.

2.The author described that PVA tape as a "water-soluble medical tape" in the manuscript. Did the author purchase specifically manufactured and officially verified tape for biomedical usage? If not, the reviewer does not think it is the correct expression.

3.The author described that the biosensor adheres to the epicardial surface by capillary adhesion force. However, the reviewer believes the bio-fluid on epicardial surface cause slippery property. Could the author clarify the mechanism of such robust coupling?

4.The reviewer thinks the author should provide continuous electrical measurement under continuous strain change to elucidate the device's feasibility for a reliable epicardial biosensor.

5.Did the author present all ECG data in this manuscript as it is without any post-filtering? If it was filtered, the author should present at least representative data before and after filtering and describe the filtering process.

6.Typically, a single electrode as a single channel is placed on the heart to obtain ECG (e.g., Xu et al., NatComm, 5, 3329 (2014)), but this work use two electrodes for a single channel. Obviously, the single-channel is better for a high-density sensor array. Why the author chose two electrodes-based sensing systems? In addition, could the author describe the sensing mechanism by two electrodes on the epicardial surface as well as the difference from a single electrode-based sensing system?

7.Since the human heart's size does not vary for each person significantly, the reviewer is concerned about the substantial usefulness of this rapid custom prototyping approach. Such devices require time to scan the heart, correspondent design, and instant fabrication without performance verification once people decide to use it. On the other hand, a pre-made device will be time-efficient since people may use it right after a decision. Besides, pre-verification of the biosensor leads to reliable function.

Point-by-Point Response Letter

We thank the reviewer #4 for constructive feedback. In the text below, we list our responses to the reviewer's comments and suggestions. The revised manuscript is enclosed.

[Comments]

Comment #1: Regarding the response to reviewer's comment #1, it is important to compare the values such as R-R interval, QRS duration, and j-point presented in Figure S12 in the revised manuscript to those by typical values from the traditional test (control), which help to give an idea whether ECG data is distorted or not during the in-vivo experiment by using the soft poroelastic biosensor.

Our Response: We thank the reviewer for this comment. For the control ECG recordings, we used commercial 3-lead electrodes (ERT Control/Gating Module Model 1030, SA Instruments, Stony Brook, NY) to obtain global (i.e., single-averaged) ECG signals across the body of a mouse. The control measurements allowed us to confirm the change of ECG waveforms over time comparing to the epicardial ECG signals simultaneously obtained from our sensors. The results showed a consistent trend of R-R interval, QRS duration, and J-point elevation between the global (control) and epicardial ECG signals. The corresponding quantitative data of R-R interval, QRS duration, and J-point elevation are $116.3 \pm 3.8/764.3 \pm 65.4$ ms, $2.8 \pm 0.5/28.8 \pm 16.1$ ms, and $-0.1 \pm 0.9/-0.1 \pm 0.4$ mV for the murine/porcine hearts, respectively (Supplementary Figure S13b). Unlike the epicardium ECG signals obtained from our sensors, the global (control) ECG signals clearly display the shift (i.e., elevation and depression) of the signal baseline caused by the inhalation and exhalation of breathing, respectively. Accordingly, we revised the following texts in the revised manuscript, “The corresponding quantitative data of R-R interval, QRS duration, and J-point elevation were measured as $116.3 \pm 3.8/764.3 \pm 65.4$ ms, $2.8 \pm 0.5/28.8 \pm 16.1$ ms, and $-0.1 \pm 0.9/-0.1 \pm 0.4$ mV for the murine/porcine hearts, respectively (Supplementary Figure S13b).” and “To confirm the change of ECG waveforms over time, Figure 4c (bottom panel) presents the results of control ECG recording that simultaneously occurred across the body of the mouse (i.e., global ECG signals) using commercial 3-lead electrodes (ERT Control/Gating Module Model 1030, SA Instruments, Stony Brook, NY). The amplitude of the global ECG signals was at least 3-fold higher than those of the epicardial ECG signals while the corresponding R-R interval, QRS duration, and J-point elevation were measured as 118.2 ± 5.7 ms, 10.0 ± 0.8 ms, and 2.0 ± 4.5 mV, respectively. Unlike the epicardium ECG signals obtained from the custom-printed sensor array, the global ECG signals obtained from the control measurement setup clearly displayed the shift (i.e., elevation and depression) of the signal baseline caused by the inhalation and exhalation of breathing, respectively.⁴⁰” in page 6.

Comment #2: Although the author has added Figure S8 for spatial mapping using 64 electrodes as responses to reviewer's comment #2 and specific comment #5, the reviewer believes that the work definitely needs spatiotemporal electrophysiological mapping results to prove the significant impact of the biosensor on cardiac physiology.

Our Response: We thank the reviewer for this comment. As the reviewer suggested, we added the corresponding spatiotemporal electrophysiological mapping results in Supplementary Figure S9 and Movie S1. Accordingly, we also added the following texts in the revised manuscript, “The corresponding spatiotemporal ECG mapping results are shown in Supplementary Figure S9 and Movie S1.” in page 5. Please note that the numbers of the figures and movies are changed in the revised manuscript.

Figure S9. Spatiotemporal ECG mapping results for the ex vivo measurement of epicardial ECG signals from the enucleated porcine heart.

Comment #3: The author emphasized the limitation of previous ink in terms of long-term reliable recording. Although the presented device itself is stable, it causes a chronic inflammatory response, as the author responds to comment #5. Therefore, this biosensor still has a long-term practical issue for implantable electronics.

Our Response: We thank the reviewer for this comment. To avoid any confusion, we removed the word of “long-term” throughout the revised manuscript. More details of the long-term inflammatory response of our sensors are discussed in the histological analysis section in page 14. In the introduction section, we also highlighted the unique features of our inks comparing to previous/existing inks. In summary, the previous/existing inks are viscoelastic and thereby limited in practical uses under periodic large strains (e.g., the heart beating) due to their mechanical and electrical hysteresis along with irreversible degradation in conductivity. On the other hand, our inks are poroelastic and thereby produced negligible mechanical and electrical hysteresis without energy loss against more than 1,000 cycles of stretching at a strain (ϵ) of 30% (Figure 2). These features of our inks are unique, which allowed us to achieve the high-fidelity acquisition of real-time 3D cardiac mapping.

Comment #4: The author mentioned anti-bio-fouling surface coating or texture as a potential solution for inflammatory response. However, the additional coating on the device causes the interlayer between the sensor and epicardium, which may cause high impedance. In addition, a textured device may also cause non-conformal contact between the sensor and epicardium. The reviewer believes the author should provide a more feasible solution.

Our Responses: We thank the reviewer for these comments. A thin anti-biofouling layer could be selectively coated across the surface of devices while the embedded recording electrodes remain uncovered [Langmuir 28, 34, 12509–12517, 2012]. In this configuration, the impedance of the devices could remain unchanged. In addition, the antifouling polymers with nano-textured surfaces could provide a strong adhesion to the wet surface of tissues in vivo owing to the hydration characteristics at the interface [Chemical Science, 11, 38, 10367-10377, 2020]. To clarify these aspects, we modified the following texts in the revised manuscript, “The inflammation could be further reduced through the inclusion of selective anti-biofouling surface coating (except for the areas of the Au recording electrodes) or the application of nanoscale texturing across the outer surface of the devices.⁵⁴⁻⁵⁶ The nano-textured surface of the devices could further improve the adhesion to the epicardium owing to the hydration characteristics at the interface.⁵⁷” in page 14.

Comment #5: Regarding specific comment #3, the author applied voltage function to the porcine heart (Figure S8). The reviewer believes those experiments may lead to the heart's electromechanical movement even though it was an ex-vivo experiment. Did the author observe such movement? If so, it will be helpful to quantify it to improve the quality of this work.

Our Response: We thank the reviewer for this question. In our measurements, we electrically stimulated the enucleated porcine heart at the applied voltage of $2 \text{ mV}\cdot\text{cm}^{-1}$ and observed no visible electromechanical movement. The electromechanical movement of the enucleated cardiac tissues have been typically observed when electrically stimulated at $> 1,000 \text{ mV}\cdot\text{cm}^{-1}$ [J. Tissue Eng. Regener. Med., 5, 6, E115–E125, 2011 & Adv. Drug Delivery Rev., 96, 135-155, 2016]. To clarify this aspect, we added the following texts in the revised manuscript, “No visible electromechanical movement of the enucleated cardiac tissues was observed throughout these measurements.” in page 5.

Comment #6: The author responded to specific comment #4 by describing Figure 5f as a spatial mapping recorded epicardial ECG from the murine heart. However, the result shows some voltage values (yellow and green) outside of the area covered by the sensor array. Could the author clarify these?

Our Response: We thank the reviewer for this question. In Figure 5f, the voltage values (yellow and green) outside of the sensor array appeared because we used an interpolation method in our post-data

processing (i.e., Python code) to smoothly fade away the colors (i.e., voltages) of the spatiotemporal 3D ECG mapping results. To clarify this aspect, we added the following texts in the revised manuscript, “The data were expressed as a matrix with n (the number of bipolar recording channels) \times time based on the spatial location of the recording electrodes and then interpolated using a custom Python code for smooth fadeaway of the colors (i.e., voltages) in the spatiotemporal 3D ECG mapping results.” in page 15.

Comment #7: The author specified, "we showed all the measurement data obtained from 5 infarcted mice (i.e., total 9 recording channels) ..." in response to the specific comment #5. Figure 5a shows 6 sensing channels on the murine heart. If the author did the same experiment with 5 different infarcted mice, the total recording channel might not be 9. The author needs to clarify.

Our Response: We thank the reviewer for this comment. For the in vivo measurements in a murine acute myocardial infarction model, a total of 3 different sensor arrays have been used on mice ($n = 5$), i.e., 1×1 ($n = 3$), 2×2 ($n = 1$), and 2×3 ($n = 1$). Therefore, the cumulative number of the recording channels is 13 through which a total of 9 infarctions (i.e., ST-elevation) were captured (Supplementary Figure S14). To avoid any confusion, we modified the following texts, “A total of 9 infarction ECG data (i.e., ST-elevation) captured from 3 different sensor arrays on mice ($n = 5$), i.e., 1×1 ($n = 3$), 2×2 ($n = 1$), and 2×3 ($n = 1$), showed consistent results as summarized in Supplementary Figure S14a, along with the corresponding quantitative data of R-R interval, QRS duration, and J-point elevation in Supplementary Figure S14b.” in page 6.

[Additional Comments]

Additional Comment #1: The author used PVA as a temporary supporting layer to deliver the sensor to the hearts. Although the author used warm saline solution to dissolve PVA, the PVA is not chemically decomposed by water, but it could be physically dissolved. Therefore, the PVA may remain just as it is in the interior of the body, which could cause some issues. Although PVA is generally considered biocompatible regarding oral intake, contact to the interior of the body and oral intake are considered different in terms of toxicity. The reviewer has concerns about it for practical usage of this work.

Our Response: We thank the reviewer for this comment. The PVA has been used for implantable medical devices (e.g., to replace cartilage and meniscus tissues) without inflammatory or pathological appearance even after a long-term implantation (> 8 months) in vivo [J. Orthop., 6, 5, 448-456, 2001 & Biomaterials, 26, 16, 3243-3248, 2005]. The biosafety of the PVA has been also proved in many other medical practices including contact lenses, hemodialysis, and synthetic vitreous humor [J. Biomed. Mater. Res. Part B: Appl. Biomater. 100B, 1451-1457, 2012]. To clarify this aspect, we added the following texts in the revised manuscript, “The water-soluble film provides excellent biocompatibility and has been used for many implantable medical devices without inflammatory responses.^{30,31}” in page 3.

Additional Comment #2: The author described that PVA tape as a "water-soluble medical tape" in the manuscript. Did the author purchase specifically manufactured and officially verified tape for biomedical usage? If not, the reviewer does not think it is the correct expression.

Our Response: We thank the reviewer for this comment. We used a commercial PVA (Polyvinyl alcohol 4-88; Sigma-Aldrich, USA) to form a water-soluble film via spin-casting or drop-casting. To clarify this aspect, we changed the term from “water-soluble medical tape” to “water-soluble film” throughout the revised manuscript.

Additional Comment #3: The author described that the biosensor adheres to the epicardial surface by capillary adhesion force. However, the reviewer believes the bio-fluid on epicardial surface cause slippery property. Could the author clarify the mechanism of such robust coupling?

Our Response: We thank the reviewer for this comment. The conformal contact of our biosensors to the epicardium is possible because of their extremely low bending stiffness ($< 8.0 \times 10^7 \text{ GPa} \cdot \mu\text{m}^4$), which results in a strong capillary adhesion (e.g., the required minimum adhesion energy per unit area $\approx 0.5 \text{ mJ} \cdot \text{m}^{-2}$) under wet environments (Figure S2b). These observations are consistent with those shown in previous reports [Nat. Mater. 9, 511–517, 2010 & Nat. Commun., 5, 1, 1-10, 2014]. To clarify this aspect, we added the following texts, “The extremely low bending stiffness ($< 8.0 \times 10^7 \text{ GPa} \cdot \mu\text{m}^4$) could substantially reduce the required minimum adhesion energy per unit area, thereby providing a strong capillary adhesion to the epicardium.^{29,32}” in page 3.

Additional Comment #4: The reviewer thinks the author should provide continuous electrical measurement under continuous strain change to elucidate the device's feasibility for a reliable epicardial biosensor.

Our Response: We thank the reviewer for this comment. Figure 2h presents the results of the continuous electrical measurement of the device with a strain of up to 30% (that corresponds to the maximum strain of the human heart). The results reveal that the normalized resistance (R/R_0) of the device remained constant below 9.0 with substantially low electrical hysteresis between 0.006 and 0.192. Figure 2i confirms that the resistance remained constant below 5.0 throughout multiple loading-unloading cycles ($> 1,000$ times) with a strain of up to 30%. To clarify this aspect, we modified the following texts in the revised manuscript, “Figure 2h presents the continuous electrical measurement of the printed device under stretching up to 30% that corresponds to the maximum strain of the human heart.³⁵ The results show that the electrical hysteresis of the printed device remained substantially low between 0.006 and 0.192 which remains at least 10-fold lower than similar counterparts reported previously.^{7,36}” and “... even after more than 1,000 cycles of stretching up to 30%.” in page 4.

Additional Comment #5: Did the author present all ECG data in this manuscript as it is without any post-filtering? If it was filtered, the author should present at least representative data before and after filtering and describe the filtering process.

Our Responses: We thank the reviewer for this comment. All the ECG data were filtered using a 5 Hz high-pass filter via the LabChart software (ADInstrument). To clarify this aspect, we included Supplementary Figure S22 to present the representative ECG data before and after the filtering. Accordingly, we also added the following texts in the Methods section, “Data processing. The ECG data were collected using a multi-channel data acquisition system (ADInstruments) at a sampling rate of 4 kHz and 1 kHz for the murine and porcine heart, respectively. The data were then filtered using a 5 Hz high-pass filter via the embedded LabChart software (Supplementary Figure S22).” in page 15. Please note that the numbers of the figures are changed in the revised manuscript.

Figure S22. Representative ECG data before (black line) and after (red line) high-pass filtering at 5 Hz.

Additional Comment #6: Typically, a single electrode as a single channel is placed on the heart to obtain ECG (e.g., Xu et al., NatComm, 5, 3329 (2014)), but this work use two electrodes for a single channel. Obviously, the single-channel is better for a high-density sensor array. Why the author chose two electrodes-based sensing systems? In addition, could the author describe the sensing mechanism by two electrodes on the epicardial surface as well as the difference from a single electrode-based sensing system?

Our Response: We thank the reviewer for this comment. In a single-electrode configuration, each recording electrode is placed on the epicardium with a single reference electrode for better space use. In a bipolar (or two electrodes) configuration, two recording electrodes are placed on the epicardium in parallel at a constant distance for better temporal resolution. In this work, we used the bipolar configuration not only to reduce common-mode noises such as power line interference to improve the signal-to-noise ratio (SNR) but also to suppress the crosstalk for high fidelity recording of ECG signals. To clarify this aspect, we added the following texts in the revised manuscript, “In this study, a bipolar recording configuration was used not only to reduce common-mode noises such as power line interference but also to suppress the crosstalk for high fidelity recording of ECG signals.³⁸” in page 5. We also changed the word from “recording channels” to “bipolar recording channels” throughout the revised manuscript.

Additional Comment #7: Since the human heart's size does not vary for each person significantly, the reviewer is concerned about the substantial usefulness of this rapid custom prototyping approach. Such devices require time to scan the heart, correspondent design, and instant fabrication without performance verification once people decide to use it. On the other hand, a pre-made device will be time-efficient since people may use it right after a decision. Besides, pre-verification of the biosensor leads to reliable function.

Our Response: We thank the reviewer for this comment. The rapid custom prototyping of the sensor arrays could be useful by exclusively covering the infarcted area of the heart rather than covering the entire area. Specifically, the overall size, geometry, and structure of the infarcted area of a specific heart were captured via non-invasive ultrasound imaging, which were used to scale, adjust, and tailor the overall layout of the sensor arrays to meet the requirement of a specific geometric accuracy (Figure 1a). As a result, the multiple recording electrodes of the sensor arrays could be precisely aligned to the

position and orientation of the infarcted area for accurate spatiotemporal ECG mapping. Moreover, the rapid custom prototyping of the sensor arrays could be also useful for spatiotemporal ECG mapping on various animal models with a wide range of heart sizes and shapes (Figure 3). In addition, the performance of the as-printed sensor arrays (e.g., electrochemical impedance) was verified on benchtop prior to their implementation on the heart in vivo (Supplementary Figure S11). As the reviewer mentioned, these pre-verifications of the as-printed sensor arrays led to reliable function. To clarify these aspects, we modified the following texts in the revised manuscript, "...with a custom fit to the infarcted area of the heart.", "...via non-invasive ultrasound imaging...", and "This custom design allowed the recording electrodes of the device to be precisely aligned to the position and orientation of the infarcted area of the heart." in page 2; "... , which could be also useful for spatiotemporal ECG mapping on various animal models with a wide range of heart sizes and shapes," in page 5; and "The electrochemical impedance of the as-printed devices was verified prior to their implementation onto the heart in vivo." in page 6.

REVIEWERS' COMMENTS

Reviewer #4 (Remarks to the Author):

The author had sufficiently addressed all the reviewer's comments. I believe the manuscript is well revised and improved. I recommend accepting this manuscript for publication in Nature Communications.